# PRIVACY-AWARE LIFELONG LEARNING

**Ozan Özdenizci**[1]**, Elmar Rueckert**[1]**, Robert Legenstein**[2]
[1] Chair of Cyber-Physical-Systems, Montanuniversität Leoben, Austria
[2] Institute of Machine Learning and Neural Computation, Graz University of Technology, Austria
`{ozan.oezdenizci,elmar.rueckert}@unileoben.ac.at`
`robert.legenstein@tugraz.at`

## ABSTRACT

Lifelong learning algorithms enable models to incrementally acquire new knowledge without forgetting previously learned information. Contrarily, the field of machine unlearning focuses on explicitly forgetting certain previous knowledge from pretrained models when requested, in order to comply with data privacy regulations on the *right-to-be-forgotten*. Enabling efficient lifelong learning with the capability to selectively unlearn sensitive information from models presents a critical and largely unaddressed challenge with contradicting objectives. We address this problem from the perspective of simultaneously *preventing catastrophic forgetting* and *allowing forward knowledge transfer* during task-incremental learning, while *ensuring exact task unlearning* and *minimizing memory requirements*, based on a single neural network model to be adapted. Our proposed solution, privacy-aware lifelong learning (PALL), involves optimization of task-specific sparse subnetworks with parameter sharing within a single architecture. We additionally utilize an episodic memory rehearsal mechanism to facilitate exact unlearning without performance degradations. We empirically demonstrate the scalability of PALL across various architectures in image classification, and provide a state-of-the-art solution that uniquely integrates lifelong learning and privacy-aware unlearning mechanisms for responsible AI applications.

## 1 INTRODUCTION

Lifelong learning algorithms enhance the ability of machine learning models to incrementally acquire new skills or integrate new knowledge over time from sequentially observed data (van de Ven et al., 2022). This continual learning capability is essential for models to stay relevant in dynamic environments where the observed data distributions change. A widely studied challenge in this setting is to mitigate *catastrophic forgetting*, addressing the loss of prior knowledge as new tasks are learned. There has been various strategies proposed to prevent forgetting, while exploiting *forward knowledge transfer* to efficiently improve performance in new tasks. However, these lifelong learning approaches conventionally do not consider the factor of ensuring data privacy, whereas selectively forgetting (or *unlearning*) certain knowledge may be required to comply with the legal regulations on the *right-to-be-forgotten* (Mantelero, 2013) (e.g., deleting prior information from personalized recommendation systems). This introduces an additional dimension of complexity, which requires novel lifelong learning solutions that can ensure unlearning for privacy-awareness.

The field of machine unlearning focuses on explicitly removing the influence of specific data points from pretrained models (Cao & Yang, 2015). Ensuring *exact unlearning*, where the model is guaranteed to behave as if the unlearned data was never observed, presents a significant challenge that generally requires partial model retraining (Bourtoule et al., 2021). In particular, current unlearning solutions assume previous or all data to be available to facilitate exact unlearning, which does not apply to lifelong learning settings where the data is only sequentially observed. Accordingly, recent works have started to explore solutions at the intersection of task-incremental lifelong learning and machine unlearning (Shibata et al., 2021; Liu et al., 2022; Chatterjee et al., 2024), primarily via inexact unlearning methods which does not guarantee privacy for all previously learned tasks.

We consider a similar lifelong learning problem, where the learning sequence may include *exact* task unlearning requests for any of the previously learned tasks, with no access to prior data. A

naive solution in this particular setting is to train independent models for each task, and discard the models corresponding to the tasks to be exactly unlearned upon request (Liu et al., 2022). However, this is inefficient since it does not enable knowledge transfer from prior tasks, and becomes memory demanding as the number of tasks increase. From a novel perspective, we present an efficient solution to this multidimensional problem by using a fixed-capacity neural network architecture.

We propose *privacy-aware lifelong learning (PALL)* as a novel framework that completely **alleviates catastrophic forgetting**, facilitates **selective knowledge transfer** from previously learned tasks, ensures **exact task unlearning guarantees** when requested, and provides a state-of-the-art solution to lifelong learning and unlearning with **minimal model memory requirements**. Our approach is based on jointly optimizing task-specific sparse subnetwork connectivity structures and their parameters within a single fixed-capacity model, and isolating this knowledge by freezing its parameters to prevent catastrophic forgetting. We facilitate learnable knowledge transfer through shared parameters by allowing this optimization process to also leverage connections with frozen weights from previous tasks, if preferred. We perform exact unlearning by resetting the subnetwork parameters that are optimized on the task to be unlearned, and use an episodic memory rehearsal mechanism to recover any performance degradation in the other tasks that may occur due to reinitialization of shared parameters which are unlearned. Our contributions are summarized as follows:

- We formulate a task-incremental learning and unlearning problem with strong privacy considerations, where exact unlearning is possible for all tasks during their lifetime.
- We present privacy-aware lifelong learning (PALL) as a memory-efficient algorithmic solution to this problem, which enables learning without catastrophic forgetting, allows learnable forward knowledge transfer, and ensures exact unlearning guarantees by design.
- We empirically demonstrate scalability of PALL on both convolutional benchmark architectures and attention-based vision transformers, yielding a stable performance in highly dynamic lifelong learning scenarios with randomly arriving unlearning requests.

## 2 RELATED WORK

### 2.1 LIFELONG LEARNING

Lifelong learning, or continual learning, explores the ability of machine learning models to adapt and learn continuously from a sequentially observed stream of data (De Lange et al., 2021; van de Ven et al., 2022; 2024). The central challenge is to address the problem of *catastrophic forgetting*, which is a widely studied phenomenon caused by traditional learning algorithms resulting in loss of previously acquired knowledge as new tasks are learned. Approaches to lifelong learning are also ideally expected to allow *forward knowledge transfer*, by leveraging information from previous tasks to enhance performance on new ones (Kudithipudi et al., 2022). Different lifelong learning scenarios are categorized as task-, class- or domain-incremental learning, which vary in terms of the target variable spaces. We focus on the task-incremental learning setting, which maintains separate label spaces for each task and assumes that the task is known by the agent. Existing approaches can be broadly divided into regularization-based, rehearsal-based, and architecture-based methods.

**Regularization-based methods**, such as elastic weight consolidation (EWC) (Kirkpatrick et al., 2017), learning without forgetting (LwF) (Li & Hoiem, 2017), synaptic intelligence (Zenke et al., 2017), and memory-aware synapses (Aljundi et al., 2018), aim to ensure that the network retains previously acquired knowledge while learning new ones by penalizing the updates to the parameters that are crucial for previously learned tasks in different ways.

**Rehearsal-based methods**, such as experience replay (Rolnick et al., 2019; Chaudhry et al., 2019) and generative modeling based rehearsal (Shin et al., 2017), store episodic training set exemplars in a buffer or use auxiliary generative models to synthesize and replay past data during training. These methods also extended to utilize gradient episodic memory (GEM) (Lopez-Paz & Ranzato, 2017), or combine replay with knowledge distillation to maintain balanced data representations (Rebuffi et al., 2017). Recently, dark experience replay (DER++) (Buzzega et al., 2020) proposed to regularize training with logit penalties to stabilize learning with respect to samples from the buffer.

**Architecture-based methods** exploit context-specific model components and reconfigure the neural network backbone structure. Progressive neural networks (Rusu et al., 2016) and dynamically

expandable neural networks (Yoon et al., 2017) adjust the model by expanding the network layers to add new capacities for new tasks when needed. To completely eliminate forgetting, the *expert gate* method duplicates the model for each new task and uses an input gating mechanism to use the relevant expert at test-time (Aljundi et al., 2017). Considering limited model memory budget settings, another line of work proposes to use distinguished sets of parameters via task-specific subnetworks within a fixed model, which are kept frozen to alleviate forgetting. PackNet (Mallya & Lazebnik, 2018) and CLNP (Golkar et al., 2019) use magnitude-based pruning to obtain these sparse subnetworks, by reusing all weights from previous tasks for knowledge transfer. Recently, methods that partially reuse the weights from previous subnetworks were developed to allow selective knowledge transfer. Specifically, Dekhovich et al. (2023) used heuristic weight importance scores for pruning based on neuron activations, and winning subnetworks (WSN) (Kang et al., 2022) employ the idea of trainable importance scores to obtain task-specific subnetworks with selective weight sharing.

## 2.2 MACHINE UNLEARNING

Machine unlearning is the process of removing the influence of specific data points from a model (Cao & Yang, 2015; Ginart et al., 2019), in order to re-establish privacy following a user's request for certain data samples, e.g., her/his own, to be deleted from the training set of the model, to comply with legal regulations on the *right-to-be-forgotten* (Mantelero, 2013). Besides updating the training set, unlearning methods modify the pretrained model to remove any influence of these samples, such that complete retraining is not needed to prevent membership inference attacks (Shokri et al., 2017).

**Exact unlearning methods** aim to completely remove the influence of targeted data, ensuring the model behaves as if the data was never observed. Beyond certified data removal from smaller scale linear models (Guo et al., 2020), exact unlearning from neural networks generally requires compute-efficient model retraining methods. The state-of-the-art approach SISA (Bourtoule et al., 2021) partitions the training set into disjoint shards and trains separate models on each shard, such that unlearning only requires retraining on affected shards. This idea was later extended to exploit data dependency structures across shards for efficiency (Dukler et al., 2023). Other examples include leveraging ensemble learning of multiple one-class tasks to reduce retraining costs (Yan et al., 2022), or minimizing parameters of the architecture for faster retraining (Yu et al., 2022).

**Approximate unlearning methods** manipulate model parameters using gradient based information to perform more efficient (but inexact) unlearning with faster retraining (Wu et al., 2020; Golatkar et al., 2020; Sekhari et al., 2021; Graves et al., 2021; Neel et al., 2021). However, such inexact solutions have been shown to require rigorous evaluations due to potentially misleading interpretations (Goel et al., 2022; Hayes et al., 2024), and involve further privacy and fairness implications for the other samples in the datasets (Chen et al., 2021; Zhang et al., 2023). Notably, approximate unlearning methods also do not generalize in a setting with *adaptive requests* (Gupta et al., 2021), which refers to the scenario where unlearning requests arrive sequentially rather than all at once.

## 2.3 SELECTIVE FORGETTING IN LIFELONG LEARNING

Unlearning in lifelong learning settings has been recently explored in the context of *beneficial forgetting* (Wang et al., 2023), as opposed to the problem of catastrophic forgetting that continual learning generally focuses on. One of the earliest methods, learning with selective forgetting (LSF) (Shibata et al., 2021), modifies models to make incorrect predictions on the unlearned data, by using auxiliary mnemonic codes to manipulate the input space. However, this only yields inexact unlearning, since poor model performance does not ensure privacy. Other works have similarly explored inexact unlearning methods, both for task-incremental learning using knowledge deposit modules (Ye et al., 2022) or student-teacher knowledge distillation mechanisms (Chatterjee et al., 2024), as well as class-incremental learning with data representation based approaches (Zuo et al., 2024).

Exact unlearning via dataset sharding and retraining (Bourtoule et al., 2021) is not applicable to lifelong learning, since there is no access to previous datasets. Recently, the continual learning and private unlearning (CLPU) framework (Liu et al., 2022) explored a related problem with another baseline approach. Specifically, CLPU defines task-incremental learning with instructions to *temporarily* or *permanently* learn the given tasks, and ensures exact unlearning on temporarily learned tasks by training independent models which are deleted upon request. In an open-world scenario with privacy guarantees on any continually learned task, this solution becomes memory inefficient.

## 3 PRIVACY-AWARE LIFELONG LEARNING (PALL)

### 3.1 PRELIMINARIES

**Lifelong Learning:** Consider a sequence of task IDs $t \in \Gamma$ where $\Gamma = \{1, \ldots, T\}$ in a supervised task-incremental learning scenario with training datasets $\mathcal{D}^t = \{(\boldsymbol{x}_1^t, y_1^t), \ldots, (\boldsymbol{x}_n^t, y_n^t)\}$, and test datasets $\mathcal{D}_{\text{test}}^t = \{(\boldsymbol{x}_1^{t,\text{test}}, y_1^{t,\text{test}}), \ldots, (\boldsymbol{x}_{n'}^{t,\text{test}}, y_{n'}^{t,\text{test}})\}$, where $\boldsymbol{x} \in \mathcal{X}$ and $y \in \mathcal{Y}^t$ denote the raw data and labels. The learner trains a neural network model $f_{\boldsymbol{\theta}}$ with parameters $\boldsymbol{\theta} \in \mathbb{R}^d$, by applying a learning algorithm $\mathcal{L}$ on $\mathcal{D}^t$ to sequentially optimize $\boldsymbol{\theta}^t \sim \mathcal{L}(\boldsymbol{\theta}^{t-1}, \mathcal{D}^t)$, often based on a cross-entropy loss $\ell_{\text{ce}}(\boldsymbol{x}, y; \boldsymbol{\theta})$, and estimates a probability distribution over $\mathcal{Y}^t$ via softmax$(f_{\boldsymbol{\theta}}(\boldsymbol{x}^t))$.

In lifelong learning, the learner loses access to $\mathcal{D}^{<t} = \{\mathcal{D}^1, \ldots, \mathcal{D}^{t-1}\}$ when learning task $t$. Moreover, for fixed model capacity, $\boldsymbol{\theta}^t$ depends on $\boldsymbol{\theta}^\tau$ for all $\tau < t$. This necessitates tailored learning algorithms to alleviate *catastrophic forgetting*, such that performance on $\mathcal{D}_{\text{test}}^{<t}$ can be maintained, while ideally achieving *forward knowledge transfer* by leveraging information from prior tasks.

**Exact Task Unlearning:** We consider a scenario where the learner is expected to *unlearn* part of the previously observed training datasets, i.e., a forget set, due to privacy related concerns. We define the forget set to be the whole training dataset $\mathcal{D}^\tau$ corresponding to a previously learned task $\tau$.[1] For a learning algorithm $\mathcal{L}$ applied to $\mathcal{D}^{\leq t}$, and a previously observed task dataset $\mathcal{D}^\tau$ to be unlearned, an *exact task unlearning* mechanism $\mathcal{U}$ uses $\boldsymbol{\theta}^t$ as a reference and returns a model such that:

$$\mathcal{U}\left(\boldsymbol{\theta}^t \sim \mathcal{L}\left(\boldsymbol{\theta}^0, \mathcal{D}^{\leq t}\right), \tau\right) =_p \mathcal{L}\left(\boldsymbol{\theta}^0, \mathcal{D}^{\leq t} \setminus \mathcal{D}^\tau\right), \tag{1}$$

where $=_p$ indicates that the models share the same probability distribution. Specifically, if the unlearned model possesses no information about $\mathcal{D}^\tau$, an adversary cannot differentiate this model from a model trained on $\mathcal{D}^{\leq t} \setminus \mathcal{D}^\tau$ from scratch based on $\mathcal{L}$, thus $\mathcal{U}$ achieves exact unlearning. In lifelong learning, this constitutes a challenging problem since there is no access to previous datasets.

### 3.2 PROBLEM STATEMENT

We formulate a generalized lifelong learning problem with privacy considerations, by extending the traditional task-incremental learning setup to allow exact task unlearning instructions. We consider that the learner receives a sequence of $r$ requests $\mathcal{R}_{1:r}$, consisting of $T$ task learning and $N_u$ task unlearning instructions which are provided in a logically consistent order (i.e., a task can only be unlearned after it has been learned). We assume that all tasks are to be learned once without repetition. The $i$-th request $\mathcal{R}_i$ in the sequence $\mathcal{R}_{1:r}$ is defined as follows:

$$\mathcal{R}_{1:r} = [\mathcal{R}_1, \mathcal{R}_2, \ldots, \mathcal{R}_r], \quad \text{such that} \quad \begin{cases} \mathcal{R}_i = (t, \mathcal{D}^t, \mathbf{L}) & \text{if task } \textit{learning}, \\ \mathcal{R}_i = (t, \mathbf{U}) & \text{if task } \textit{unlearning}, \end{cases} \tag{2}$$

where $\mathbf{L}$ and $\mathbf{U}$ are flag variables to indicate if the instruction corresponds to a learning or unlearning request. Furthermore, the learner keeps a dictionary $\Omega_i$ of the currently learned task IDs that were not unlearned: $\Omega_i \leftarrow \Omega_{i-1} \cup \{t\}$ if learning task $t$, and $\Omega_i \leftarrow \Omega_{i-1} \setminus \{t\}$ if unlearning task $t$.

The learner's goal is to solve this problem by defining a learning algorithm $\mathcal{L}$, and an unlearning algorithm $\mathcal{U}$, to be applied sequentially based on $\mathcal{R}_i$ to optimize the model parameters to achieve:

$$\boldsymbol{\theta}^i \sim \begin{cases} \mathcal{L}\left(\boldsymbol{\theta}^{i-1}, \mathcal{D}^t\right) \;\; \text{s.t.} \;\; \min_{\boldsymbol{\theta}} \dfrac{1}{|\Omega_i|} \sum_{t \in \Omega_i} \mathbb{E}_{(\boldsymbol{x}, y) \sim \mathcal{D}_{\text{test}}^t}\left[\ell_{\text{ce}}(\boldsymbol{x}, y; \boldsymbol{\theta})\right] & \text{if } \mathcal{R}_i = (t, \mathcal{D}^t, \mathbf{L}), \\[2ex] \mathcal{U}\left(\boldsymbol{\theta}^{i-1}, t\right) \;\;\; \text{s.t.} \;\; \mathcal{U}\left(\boldsymbol{\theta}^{i-1}, t\right) =_p \mathcal{L}\left(\boldsymbol{\theta}^0, \mathcal{D}^{[\tau \in \Omega_i]}\right) & \text{if } \mathcal{R}_i = (t, \mathbf{U}). \end{cases} \tag{3}$$

A holistic solution to this problem would **mitigate catastrophic forgetting** as new tasks are learned, **allow forward knowledge transfer** for efficient learning, **ensure privacy-awareness** with exact unlearning guarantees, and **minimize memory requirements** of the algorithm. Our formulation differs from the CLPU (Liu et al., 2022) setting by generalizing the problem in terms of its privacy constraints such that *any* task can always be exactly unlearned (i.e., all tasks are temporarily learned).

---

[1]Differently from traditional machine unlearning studies that generally define the forget set to be a subset of training samples or a certain class in the training dataset, we focus on whole task unlearning scenarios.

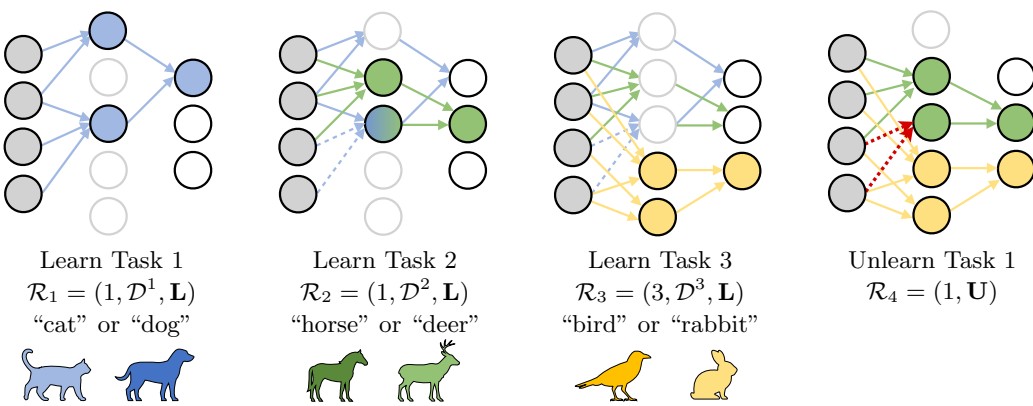

Figure 1: Illustration of PALL. Task-specific subnetworks obtained after learning are indicated by color. The subnetwork mask $\mathbf{m}_2$ for Task 2 contains two shared, frozen parameters from Task 1 (dashed blue lines), as well as $\overline{\mathbf{m}}_2$ (green connections). Following the unlearning request for Task 1, we reset all parameters trained on $\mathcal{D}^1$ (blue connections), and retrain any of those parameters which were used for knowledge transfer in later tasks (shown by red connections) using experience replay.

### 3.3 EFFICIENT LIFELONG LEARNING WITH EXACT TASK UNLEARNING

Existing lifelong learning methods are not designed for exact task unlearning capabilities. Recent studies have only explored inexact unlearning methods in this context (Shibata et al., 2021; Chatterjee et al., 2024). A naive solution to satisfy Eq. (3) would be to train independent models for each task, where one can ensure exact unlearning by deleting the task-specific model upon request.[2] However, this approach becomes infeasible under limited memory as the number of tasks increases.

We propose **privacy-aware lifelong learning (PALL)** as a memory-efficient hybrid solution that utilizes an architecture-based lifelong learning approach, combined with an episodic memory rehearsal mechanism. We optimize task-specific subnetworks within a single architecture with *limited model memory*, where the associated parameters are kept frozen to *eliminate catastrophic forgetting*. During task-specific subnetwork optimization, we *allow learnable knowledge transfer* to future tasks by selectively reusing parameters from previous task subnetworks (Kang et al., 2022). We ensure *exact task unlearning* by resetting the associated task subnetwork upon request, and use experience replay to mitigate potential performance degradations in the other tasks that may occur due to the reinitialization of shared parameters. PALL is illustrated in Figure 1 and described in detail below.

**Given a task learning request** $\mathcal{R}_i = (t, \mathcal{D}^t, \mathbf{L})$, our goal is to optimize a sparse subset of the current parameters $\boldsymbol{\theta}^{i-1}$, and a binary subnetwork mask $\mathbf{m}_t \in \{0, 1\}^d$, which will be used for inference on task $t$. We perform this by jointly optimizing the parameters that are *unused* in previous tasks, and task-specific importance scores $\mathbf{s}_t$, which quantifies the significance of each parameter:

$$\min_{\boldsymbol{\theta}, \mathbf{s}_t} \mathbb{E}_{(\boldsymbol{x}, y) \sim \mathcal{D}^t} \left[ \ell_{\text{ce}} \left( \boldsymbol{x}, y; \boldsymbol{\theta} \odot \mathbf{m}_t(\mathbf{s}_t) \right) \right]. \tag{4}$$

We compute $\mathbf{m}_t$ at each iteration using the current largest $|\mathbf{s}_t|$ values on a per layer basis, based on a connectivity rate $\alpha$ (e.g., $\alpha = 0.1$ indicates 90% sparsity) (Ramanujan et al., 2020). Since we do not want to change the parameters trained on previous tasks, we perform masking of parameter updates:

$$\boldsymbol{\theta} \leftarrow \boldsymbol{\theta} - \eta \left( \frac{\partial \ell_{\text{ce}}}{\partial \boldsymbol{\theta}} \odot (1 - \mathbf{M}_{i-1}) \right), \qquad \mathbf{s}_t \leftarrow \mathbf{s}_t - \eta \left( \frac{\partial \ell_{\text{ce}}}{\partial \mathbf{s}_t} \right), \tag{5}$$

where $\mathbf{M}_{i-1} = \bigvee_{j \in \Omega_{i-1}} \mathbf{m}_j$ denotes the *cumulative binary mask* which identifies the combined set of used and frozen subnetwork parameters until the $i$-th request. The scores $\mathbf{s}_t$ are optimized via a straight-through estimator on the binarizing mask $\mathbf{m}_t(\mathbf{s}_t)$ during backpropagation.

This objective allows *learnable forward knowledge transfer* by optimizing $\mathbf{s}_t$ for all parameters of the model without masking, such that $\mathbf{m}_t$ for different tasks can be overlapping to share parame-

---

[2]This is identical to the recently proposed CLPU solution (Liu et al., 2022) in our experimental setting that requires exact unlearning guarantees for any continually learned task.

ters. We indicate the parameter indices which are specifically trained using data from $\mathcal{D}^t$ with the submask $\overline{\mathbf{m}}_t$, and $\mathbf{m}_t - \overline{\mathbf{m}}_t$ correspond to the shared, frozen parameter indices from previous tasks. After task learning, we discard $\mathbf{s}_t$ and store the final $\mathbf{m}_t$ in a dictionary of binary subnetwork masks $\mathcal{M}_i = \{\mathbf{m}_j \mid j \in \Omega_i\}$. Due to the parameter masking strategy used during training, we can always use the corresponding $\mathbf{m}_t$ to make consistent predictions on $\mathcal{D}_{\text{test}}^t$ without catastrophic forgetting.

Importantly, we update $\mathbf{M}_i \leftarrow \mathbf{M}_{i-1} \vee \mathbf{m}_t$ and reset all unused parameters $\boldsymbol{\theta}^i \odot (1 - \mathbf{M}_i)$ by sampling from the weight initialization distribution $\phi(.)$ after training. This ensures that no information from $\mathcal{D}^t$ leaks into the remaining unused parameters outside the ones identified by $\mathbf{m}_t$, and helps to ensure future unlearning guarantees. Lastly, we store a set of randomly sampled exemplars and logits to an episodic memory buffer $\mathcal{B}^t = \{(\boldsymbol{x}_j, y_j, \boldsymbol{z}_j = f_{\boldsymbol{\theta}^i \odot \mathbf{m}_t}(\boldsymbol{x}_j)) \mid (\boldsymbol{x}_j, y_j) \sim \mathcal{D}^t\}_{1 \leq j \leq |\mathcal{B}^t|}$. We do not use this buffer for task learning, but will use these samples upon unlearning requests.

**Given a task unlearning request** $\mathcal{R}_i = (t, \mathbf{U})$, our goal is to update the model parameters $\boldsymbol{\theta}^{i-1}$, such that the new model does not possess any information about $\mathcal{D}^t$, i.e., none of its parameters have been optimized with the data observed from task $t$. We can facilitate this exactly by resetting the parameters $\boldsymbol{\theta}^{i-1} \odot \overline{\mathbf{m}}_t$, by sampling new values from the initialization distribution $\phi(.)$.

If the unlearning request $\mathcal{R}_i = (t, \mathbf{U})$ refers to the latest task that was learned in $\mathcal{R}_{i-1} = (t, \mathcal{D}^t, \mathbf{L})$, we can simply rewind this learning instruction by resetting $\boldsymbol{\theta}^{i-1} \odot \overline{\mathbf{m}}_t$. However, if the unlearning request refers to an earlier task $t$ which was followed by other task learning requests $\tau > t$ and $\tau \in \Omega_i$, then purely resetting $\boldsymbol{\theta}^{i-1} \odot \overline{\mathbf{m}}_t$ will lead to a performance degradation for tasks $\tau$, if $\mathbf{m}_\tau$ is overlapping with $\overline{\mathbf{m}}_t$ to share parameters from task $t$ (red connections in Figure 1). To address this conflict between knowledge transfer and exact unlearning, we use memory buffer rehearsal and perform a short *retraining* step on such affected parameters following the objective:

$$\min_{\overline{\boldsymbol{\theta}}} \sum_{\substack{\tau > t \\ \tau \in \Omega_i}} \frac{1}{|\mathcal{B}^\tau|} \left[ \sum_{(\boldsymbol{x}, y, \boldsymbol{z}) \sim \mathcal{B}^\tau} \ell_{\text{ce}}\left(\boldsymbol{x}, y; \boldsymbol{\theta} \odot \mathbf{m}_\tau\right) + \beta \cdot \sum_{(\boldsymbol{x}', y', \boldsymbol{z}') \sim \mathcal{B}^\tau} ||f_{\boldsymbol{\theta} \odot \mathbf{m}_\tau}(\boldsymbol{x}') - \boldsymbol{z}'||_2^2 \right], \quad (6)$$

where $\overline{\boldsymbol{\theta}}$ denotes the affected subset of parameters within $\boldsymbol{\theta}$ that were reset, which are indicated by $\bigvee_{\tau > t, \tau \in \Omega_i}(\mathbf{m}_\tau \wedge \overline{\mathbf{m}}_t)$. Eq. (6) is a generalized formulation used in various rehearsal based lifelong learning methods, where $\beta = 0$ would yield vanilla experience replay (Chaudhry et al., 2019) and $\beta = 0.5$ yields DER++ (Buzzega et al., 2020). We perform $N_f$ retraining iterations for Eq. (6). Finally, we delete $\mathcal{B}^t$ and $\mathbf{m}_t$, and re-compute $\mathbf{M}_i = \bigvee_{j \in \Omega_i} \mathbf{m}_j$. Our algorithm is in Appendix A.2.

## 4 EXPERIMENTAL SETUP

### 4.1 DATASETS AND MODELS

We performed experiments with sequential CIFAR10 (S-CIFAR10: 5 tasks $\times$ 2 classes), CIFAR100 (S-CIFAR100: 10 tasks $\times$ 10 classes) and TinyImageNet (S-TinyImageNet: 20 tasks $\times$ 10 classes, 40 tasks $\times$ 5 classes, or 100 tasks $\times$ 2 classes) datasets. We used ResNet-18 and ResNet-34 models in S-CIFAR10/100 experiments which are commonly used as benchmarks in lifelong learning, and attention-based ViT-T/8 architectures with S-TinyImageNet (see Appendix A.1 for details).

We designed lifelong learning scenarios with $T$ task *learning* instructions, and $N_u$ randomly chosen task *unlearning* instructions, which are arranged in a logically consistent manner, e.g., a user request sequence on S-CIFAR10 with $N_u = 3$ can be: $\mathcal{R}_{1:8} = [(1, \mathcal{D}^1, \mathbf{L}), (2, \mathcal{D}^2, \mathbf{L}), (3, \mathcal{D}^3, \mathbf{L}), (2, \mathbf{U}), (4, \mathcal{D}^4, \mathbf{L}), (3, \mathbf{U}), (5, \mathcal{D}^5, \mathbf{L}), (1, \mathbf{U})]$. Experiments are repeated using 20 random seeds (unless stated otherwise) for a given $N_u$, which results in randomly changing the allocation of the classes into different tasks, as well as the unlearning instructions and their order within $\mathcal{R}_{1:r}$.

### 4.2 MODEL TRAINING AND EVALUATIONS

**Baseline Methods:** We compare our results against state-of-the-art lifelong learning baselines: sequential learning by directly finetuning the model on each new task (Sequential), elastic weight consolidation (EWC) (Kirkpatrick et al., 2017), learning without forgetting (LwF) (Li & Hoiem, 2017), learning with selective forgetting (LSF) (Shibata et al., 2021), gradient episodic memory

(GEM) (Lopez-Paz & Ranzato, 2017), experience replay (ER) (Chaudhry et al., 2019), dark experience replay (DER++) (Buzzega et al., 2020), PackNet (Mallya & Lazebnik, 2018), winning subnetworks (WSN) (Kang et al., 2022), and task-specific independent model training (Independent) which is equivalent to the naive solution by CLPU (Liu et al., 2022) in our problem setting.

These baseline approaches, except for LSF (Shibata et al., 2021) and Independent (Liu et al., 2022), are not originally designed with task unlearning capabilities. Thus, we adapt these methods to the current problem. Particularly for GEM, ER and DER++, for task unlearning, we perform finetuning for $N_f$ iterations on the remaining episodic memories and predict uniform distributions using the unlearned task's episodic memories to accelerate forgetting, prior to removing the corresponding episodic memory of the unlearned task. For Sequential, EWC, LwF, PackNet and WSN, we do not perform any changes to the model parameters for task unlearning. We only discard the algorithm-specific stored variables associated with the task to be unlearned, e.g., the subnetwork masks in PackNet and WSN (see "Unlearning Implementations" under Appendix A.2 for further details).

**Training Configurations:** We use a stochastic gradient descent (SGD) optimizer with momentum for 20 epochs per S-CIFAR10/100 task learning instruction, with a batch size of 32, learning rate of 0.01, and weight decay with parameter 0.0005. For S-TinyImageNet, we use an Adam optimizer for 100 epochs with a batch size of 256, and a cosine annealing learning rate scheduler with an initial value of 0.001. Here, we do not use weight decay but instead apply dropout to intermediate activations of ViT-T/8 with $p = 0.1$ (Steiner et al., 2022). All methods requiring a memory buffer had a total capacity of 500 and 1000 samples (evenly split across tasks) in S-CIFAR and S-TinyImageNet experiments, respectively. Unless otherwise specified, for Eq. (6) we use $N_f = 50$ and $\beta = 0.5$ (see Appendix A.2 for further details). Our code is available at: https://github.com/oozdenizci/PALL.

**Evaluation Metrics:** We evaluate average test set accuracies for the remaining learned tasks after processing $\mathcal{R}_{1:r}$, i.e., tasks in the set $\Omega_r$, and the average test set accuracies for the unlearned tasks after processing $\mathcal{R}_{1:r}$, i.e., tasks in the set $\Gamma \setminus \Omega_r$, denoted as $\mathcal{A}_l$ and $\mathcal{A}_u$ as follows:

$$\mathcal{A}_l = \frac{1}{|\Omega_r|} \sum_{t \in \Omega_r} a_{r,t}, \qquad \mathcal{A}_u = \frac{1}{N_u} \sum_{t \in \Gamma \setminus \Omega_r} a_{r,t}, \tag{7}$$

where $a_{i,t}$ denotes the accuracy on $\mathcal{D}_{\text{test}}^t$ after request $i$ was completed. We expect better privacy-aware lifelong learning methods to have higher $\mathcal{A}_l$, and chance-level $\mathcal{A}_u$ by performing random classification on unlearned tasks. However, it is important to note that a lower $\mathcal{A}_u$ does not necessarily correspond to an exact unlearning guarantee. We include $\mathcal{A}_u$ only to evaluate inexact unlearning baselines through a weak measure. We leave detailed investigation of inexact unlearning methods with better metrics, e.g., via empirical privacy auditing (Steinke et al., 2024), for future work.

We evaluate the forgetting impact of task learning and unlearning requests, similar to the notion of backward knowledge transfer in standard continual learning. Specifically, we define $\mathcal{F}_l$ and $\mathcal{F}_u$ by evaluating the average decrease in the test set performance for previously learned tasks, after processing a task learning or unlearning request, which are formally defined as:

$$\mathcal{F}_l = \frac{1}{T-1} \sum_{\substack{i \in \{2,\ldots,r\} \\ \mathcal{R}_i = (-,\mathbf{L})}} \sum_{t \in \Omega_{i-1}} \frac{(a_{i-1,t} - a_{i,t})}{|\Omega_{i-1}|}, \quad \mathcal{F}_u = \frac{1}{N_u} \sum_{\substack{i \in \{2,\ldots,r\} \\ \mathcal{R}_i = (-,\mathbf{U})}} \sum_{t \in \Omega_i} \frac{(a_{i-1,t} - a_{i,t})}{|\Omega_i|}. \tag{8}$$

We expect better privacy-aware lifelong learning methods to have lower $\mathcal{F}_l$ and $\mathcal{F}_u$ such that there is no degrading backward transfer impact of learning or unlearning requests.

# 5 EXPERIMENTAL RESULTS

## 5.1 COMPARISONS TO STATE-OF-THE-ART IN LIFELONG LEARNING

In Table 1 we evaluate our approach against state-of-the-art methods in lifelong learning, by extending various methods to the experimental setting of task incremental learning and unlearning. We consider independent model training for each task as an upper bound baseline with exact unlearning, which however requires a model size that linearly scales with the number of tasks for inference. Our method, PALL, provides a novel, state-of-the-art solution in a privacy-aware continual learning and unlearning setting, considering all four metrics together with model memory requirements.

Table 1: Evaluations across different datasets and models. In this experimental setting, using independent models (bottom row) is identical to CLPU (Liu et al., 2022). Methods with *exact* unlearning perform random classification on unlearned tasks ($\mathcal{A}_u$). Results are averaged over 20 random seeds, where the sequence of requests are randomly generated with $N_u = 3$ unlearning instructions (see Appendix A.3.5 for worst-case results across seeds). $\alpha$: task-specific subnetwork connectivity rate.

| | **S-CIFAR10** ($T = 5$) | | | | **S-CIFAR100** ($T = 10$) | | | | **S-TinyImageNet** ($T = 20$) | | | | Model Size |
| | $\mathcal{A}_l \uparrow$ | $\mathcal{A}_u \downarrow$ | $\mathcal{F}_l \downarrow$ | $\mathcal{F}_u \downarrow$ | $\mathcal{A}_l \uparrow$ | $\mathcal{A}_u \downarrow$ | $\mathcal{F}_l \downarrow$ | $\mathcal{F}_u \downarrow$ | $\mathcal{A}_l \uparrow$ | $\mathcal{A}_u \downarrow$ | $\mathcal{F}_l \downarrow$ | $\mathcal{F}_u \downarrow$ | (Inference) |
|---|---|---|---|---|---|---|---|---|---|---|---|---|---|
| Sequential | 70.71 | 72.65 | 13.86 | 0.0 | 35.35 | 40.07 | 13.43 | 0.0 | 19.41 | 21.42 | 7.65 | 0.0 | |
| EWC | 74.27 | 73.28 | 12.02 | 0.0 | 56.01 | 52.04 | 7.03 | 0.0 | 54.74 | 53.63 | 0.49 | 0.0 | |
| LwF | 91.65 | 86.99 | 1.54 | 0.0 | 58.83 | 63.94 | 4.41 | 0.0 | 43.88 | 49.53 | 2.10 | 0.0 | $d$ |
| LSF | 89.25 | 80.25 | 0.36 | 1.26 | 56.59 | 52.88 | 1.67 | 4.93 | 43.40 | 44.88 | 0.94 | 4.60 | |
| GEM | 87.70 | 54.14 | 4.10 | 1.28 | 57.44 | 42.80 | 6.50 | 3.76 | 42.62 | 29.32 | 4.32 | 1.11 | |
| ER | 87.88 | 58.48 | 3.45 | 1.61 | 57.63 | 42.69 | 4.67 | 7.91 | 42.30 | 28.02 | 4.44 | 0.64 | |
| DER++ | 92.04 | 53.50 | 1.62 | 0.66 | 66.84 | 46.56 | 4.52 | 0.95 | 46.50 | 34.65 | 3.78 | 0.43 | |
| PackNet | 94.77 | 75.76 | **0.0** | 0.0 | 75.01 | 58.19 | **0.0** | 0.0 | 60.50 | 50.72 | **0.0** | 0.0 | |
| WSN | 94.15 | 74.76 | **0.0** | 0.0 | 73.64 | 51.52 | **0.0** | 0.0 | 63.67 | 15.16 | **0.0** | 0.0 | $d + \mathcal{M}_i$ |
| **PALL** ($\alpha = 0.05$) | 94.01 | *Exact* | **0.0** | 0.30 | 70.60 | *Exact* | **0.0** | 0.51 | 62.14 | *Exact* | **0.0** | 0.64 | |
| **PALL** ($\alpha = 0.1$) | 94.50 | *Exact* | **0.0** | 0.24 | 72.35 | *Exact* | **0.0** | 0.40 | 61.36 | *Exact* | **0.0** | 0.72 | |
| **PALL** ($\alpha = 0.2$) | 94.34 | *Exact* | **0.0** | 0.60 | 72.50 | *Exact* | **0.0** | 1.10 | 61.11 | *Exact* | **0.0** | 0.91 | |
| Independent | 95.19 | *Exact* | **0.0** | 0.0 | 73.22 | *Exact* | **0.0** | 0.0 | 61.69 | *Exact* | **0.0** | 0.0 | $d \times |\Omega_i|$ |

Regularization-based methods EWC and LwF, as well as sequential training, were indifferent to task unlearning instructions, since the original methods are not adapted to unlearning (i.e., $\mathcal{A}_u \approx \mathcal{A}_l$ and $\mathcal{F}_u = 0.0$). We observed LSF to strongly mitigate catastrophic forgetting (low $\mathcal{F}_l$), but its use of mnemonic codes (Shibata et al., 2021) for finetuning during unlearning was ineffective in our larger scale problems (i.e., above chance-level $\mathcal{A}_u$). Rehearsal based finetuning for unlearning with GEM, ER and DER++ resulted in better, lower $\mathcal{A}_u$ metrics closer to chance-levels. However, this is still an inexact unlearning approach, and all three methods still minimally suffer from catastrophic forgetting ($\mathcal{F}_l > 0$). Architecture-based methods PackNet and WSN mitigate catastrophic forgetting with frozen parameters ($\mathcal{F}_l = 0$), while only increasing the model size with $\mathcal{M}_i$, similar to PALL. PackNet and WSN also achieve $\mathcal{F}_u = 0$, since unlearning involves deletion of the corresponding mask without any change to the parameters. However, this makes unlearning inexact, since the parameters trained on the unlearned task remain. Generally, PackNet and WSN was found to perform well in task learning ($\mathcal{A}_l$), since they are not affected by parameter resetting in unlearning (e.g., PackNet: 75.01, WSN: 73.64, PALL ($\alpha = 0.2$): 72.50, Independent: 73.22 on S-CIFAR100).

Our method satisfies exact unlearning (i.e., random classification $\mathcal{A}_u$ on unlearned tasks), no catastrophic forgetting ($\mathcal{F}_l = 0$), and achieves $\mathcal{A}_l$ metrics very close to, or higher than training independent models with exact unlearning guarantees (e.g., Independent: 61.69, PALL ($\alpha = 0.05$): 62.14 on S-TinyImageNet). Moreover, rehearsal-based retraining of the reset parameters yields relatively low $\mathcal{F}_u$ (∼below 1%), which shows the efficiency of the designed unlearning process.

Henceforth, we consider $\alpha = 1/T$ for PALL, which is determined by the experimental setting. If the number of tasks to be learned are not known a priori and $\alpha > 1/T$, PALL will still allow learning via knowledge transfer from frozen weights, until some tasks are unlearned to free trainable parameters.

**Memory Requirements of PALL:** Our method requires minimal memory overhead in the total model size for inference, by partitioning multiple tasks within a limited number of floating-point parameters. To achieve this, PALL stores an additional binary mask dictionary $\mathcal{M}_i$, which includes at most $T$ masks to perform inference. This indicates $d$ parameters (32-bits), and $\mathcal{M}_i = \{\mathbf{m}_j\}_{j \in \Omega_i}$ with $d$-dimensional boolean (1-bit) masks. Alternatively, training independent models for each task can reach to a maximum of $d \times T$ parameters (32-bits), in a scenario where all tasks are learned without unlearning. Therefore, between the two existing methods for lifelong learning with exact unlearning capabilities, PALL becomes the memory-efficient choice.

Specifically, our default ResNet-18 with $d = 11.2M$ parameters on S-CIFAR10, ResNet-34 with $d = 21.3M$ on S-CIFAR100, and ViT-T/8 models with $d = 5.4M$ on S-TinyImageNet, represented in 32-bits had model sizes of 42.59 MB, 81.30 MB, and 20.63 MB, respectively. For PALL, as well as PackNet and WSN, the maximum model size that can be achieved where $|\Omega_r| = T$, yielded model sizes of 49.24 MB, 106.70 MB, and 34.41 MB, respectively, considering $T$ additional binary masks for each layer. In the case of independent models, this scenario yields total model sizes of

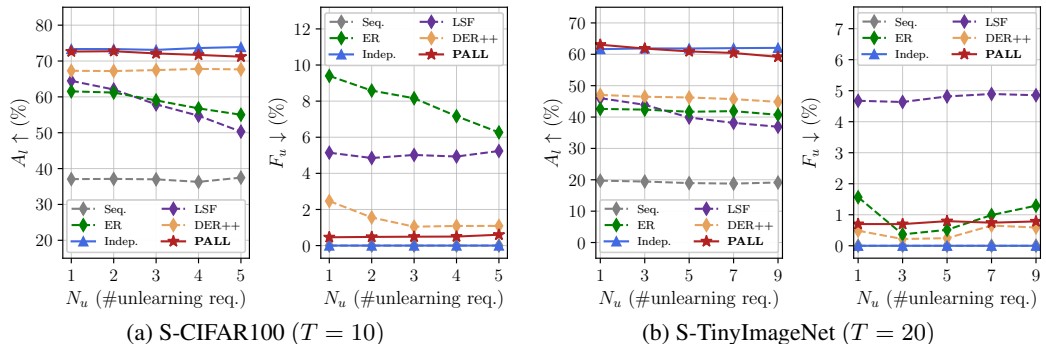

Figure 2: Evaluations with different number of unlearning requests $N_u$ in $\mathcal{R}_{1:r}$ (averaged across 10 random seeds each), for the methods that perform retraining or finetuning after task unlearning. We excluded GEM since the performance was similar to ER. We use $\alpha = 1/T$ for PALL.

212.95 MB, 813.0 MB, and 412.6 MB, respectively, indicating that PALL provides approximately $4.3\times$, $7.6\times$ and $12\times$ more model size efficient solutions with exact unlearning.

Notably, to facilitate model updates, each method also involves additional storage requirements (e.g., previous model weights in EWC, mnemonic codes in LSF). Similarly, PALL additionally requires the memory buffers $\{\mathcal{B}^t\}_{t \in \Omega_i}$, which is also common in all rehearsal-based learning methods. We further compare and discuss training times associated with each algorithm in Appendix A.3.4.

## 5.2 EMPIRICAL ANALYSIS OF THE TASK UNLEARNING MECHANISM

In Figure 2 we demonstrate the impact of $N_u$, on the methods with retraining or finetuning instructions during task unlearning. We specifically aim to assess the performance of our algorithm in task-incremental lifelong learning scenarios with more frequent unlearning requests.

We observed that PALL yields a relatively stable performance close to the naive baseline of training independent models without memory-efficiency considerations (red vs. blue solid lines in Figure 2). Particularly for large $N_u$, PALL shows less performance degradation with a parameter reset and retraining mechanism, than alternative inexact unlearning methods with finetuning: $\mathcal{A}_l \uparrow / \mathcal{F}_u \downarrow$ on S-CIFAR100 at $N_u = 5$: LSF: 50.3/5.2, DER++: 67.7/1.1, PALL: 71.2/0.6, Indep.: 73.9/0.0, and on S-TinyImageNet at $N_u = 9$: LSF: 36.9/4.9, DER++: 44.8/0.6, PALL: 59.2/0.7, Indep.: 62.1/0.0. We present additional experiments on the scalability of PALL in longer lifelong learning scenarios with S-TinyImageNet ($T = 100$) and even larger $N_u$, in Table A6 of Appendix A.3.3.

**Impact of Retraining After Unlearning:** We investigate the impact of $N_f$ during rehearsal-based retraining of the reset parameters in Appendix A.3.1. Mainly, our results show that simply resetting the affected parameters without retraining yields comparably worse performance (e.g., $\mathcal{A}_l$ for S-CIFAR100 with $N_f = 0$: 70.24 vs $N_f = 50$: 72.35), indicating the necessity of retraining the reset weights through a memory buffer. We were also able to achieve better performance recovery after unlearning by using longer retraining durations (e.g., $\mathcal{A}_l$ for S-CIFAR100 with $N_f = 100$: 72.46).

We also present results on the ratio of retrained parameters during unlearning, and obtained parameter values after retraining. We observed that only $1-4\%$ of the parameters needed to be retrained via Eq. (6), resulting in weights numerically different from those before unlearning.

**Influence of Episodic Memory Rehearsal:** We performed ablation experiments on our choices for the episodic memory rehearsal method in Appendix A.3.2. In brief, we observed that $\beta = 0.5$ is the preferable choice for retraining, as previously claimed by DER++ (Buzzega et al., 2020), and using a larger memory buffer size generally increases performance (e.g., S-CIFAR100, $\mathcal{A}_l \uparrow / \mathcal{F}_u \downarrow$ with buffer size 200: 71.90/0.72, buffer size 500: 72.35/0.40, buffer size 1000: 73.58/0.23).

We also investigate the influence of the random exemplar sampling method used to select the samples to be stored in the memory buffer in Appendix A.3.2. We observed that using prioritized sampling mechanisms (Rebuffi et al., 2017) that are different than random, did not improve performance.

## 5.3 COMPARISONS TO INDEPENDENT SUBNETWORKS WITHOUT KNOWLEDGE TRANSFER

We designed an ablation experiment where we evaluate independently trained smaller architectures with an equivalent total model size to PALL, i.e., using models with $d/T$ parameters. Similar to our baseline *Independent*, this setting also ensures exact unlearning, no catastrophic forgetting ($\mathcal{F}_l = 0$), and no impact of unlearning ($\mathcal{F}_u = 0$). We consider two configurations: (1) *static sparsity*, where $T$ *independent* sparse subnetworks with $1/T$ connectivity are randomly initialized within a model, (2) *dynamic sparsity*, where we also optimize the sparse connectivity structure of these $T$ *independent* subnetworks via score optimization. The latter, i.e., independent models via dynamic sparsity, resembles to PALL with the only difference of not allowing knowledge transfer across subnetworks via weight sharing. This also eliminates the need for a memory buffer and parameter retraining for exact unlearning.

In Table 2, we present our results on S-TinyImageNet with 40 or 100 tasks, where the architecture is divided into very small, task-specific

Table 2: Comparisons on S-TinyImageNet (40 tasks × 5 classes, and 100 tasks × 2 classes), with independent subnetworks at $1/T$ sparsity. Results are averaged over 10 random seeds with $N_u = 3$. PALL uses $N_f = 10$ retraining iterations.

| | $T = 40$ | | $T = 100$ | | Model Size |
|---|---|---|---|---|---|
| | $\mathcal{A}_l \uparrow$ | $\mathcal{F}_u \downarrow$ | $\mathcal{A}_l \uparrow$ | $\mathcal{F}_u \downarrow$ | (Inference) |
| Static Sparse (Ind.) | 70.30 | 0.0 | 80.70 | 0.0 | $d + \mathcal{M}_i$ |
| Dynamic Sparse (Ind.) | 71.19 | 0.0 | 83.75 | 0.0 | $d + \mathcal{M}_i$ |
| **PALL** ($\alpha = 0.01$) | 71.00 | 0.56 | 85.89 | 0.53 | $d + \mathcal{M}_i$ |
| **PALL** ($\alpha = 0.025$) | 72.07 | 0.36 | 86.11 | 0.43 | $d + \mathcal{M}_i$ |
| **PALL** ($\alpha = 0.05$) | 72.03 | 0.32 | 85.80 | 0.35 | $d + \mathcal{M}_i$ |
| Independent | 71.76 | 0.0 | 86.80 | 0.0 | $d \times |\Omega_i|$ |

subnetworks (e.g., 54K params at 99% sparsity), and learning without knowledge transfer becomes challenging. We observed that dynamic sparsity outperforms independent subnetworks with static sparsity, and PALL consistently outperforms all independent subnetworks with the use of knowledge transfer, e.g., for $T = 100$, PALL ($\alpha = 0.025$): 86.11, Dynamic: 83.75. This makes PALL favorable in longer lifelong learning scenarios, where the number of tasks can be very high or unknown.

## 6 DISCUSSION

Lifelong learning and machine unlearning explores two important, yet mostly independently studied aspects of truly adaptive, flexible, and responsible AI systems. We proposed PALL as an algorithmic solution combining these two challenging and contradicting aspects, by satisfying all key pillars in both domains (i.e., no catastrophic forgetting, forward knowledge transfer, exact unlearning guarantees, memory-efficiency), for the first time. Our empirical evaluations demonstrate the effectiveness and scalability of PALL in dynamic environments, where efficient task learning and exact unlearning capabilities are desired by state-of-the-art models. Notably, we have shown PALL to yield up to $12\times$ more model size efficient solutions with better or comparable task learning performances, as opposed to the naive baseline of training independent models to satisfy exact unlearning.

Our method is partially based on architecture-based lifelong learning methods that allow selective knowledge transfer (Kang et al., 2022; Ramanujan et al., 2020), which we innovatively extended into a hybrid learning strategy that is also equipped with a rehearsal-based lifelong learning method (Buzzega et al., 2020). This enables our algorithm to handle exact task unlearning requests in the presence of knowledge transfer, which was not addressed to date. Additionally, we also performed critical modifications to the existing subnetwork optimization methods, such as reinitializing the scores and unused weights after each learning request. These were required to satisfy overall exact unlearning guarantees, and resulted in effective learning and unlearning abilities simultaneously.

In this work, we focus on task-incremental learning settings, where the task ID is available to the learner. Our approach is not yet readily applicable to a class- or domain-incremental scenario where all previously learned tasks' label spaces are unified, since PALL disentangles the choice of the task-specific subnetwork based on the task IDs to ensure exact unlearning. To be applicable in these settings, PALL can be extended with a privacy-aware auxiliary algorithm to first identify the task, and subsequently utilize task-specific subnetwork via gating (Aljundi et al., 2017; Von Oswald et al., 2019). Our work also does not yet consider selective data unlearning, but instead performs complete task unlearning. To achieve stricter privacy with deletion guarantees for each data sample, our approach can be combined with differentially private optimization methods (Lai et al., 2022), in future work. Finally, going beyond our scope on vision tasks, we believe that applying PALL to language processing tasks in future work would also be of broad interest for responsible AI systems.

REPRODUCIBILITY STATEMENT

We provide detailed descriptions of the training configurations and hyperparameters of the experiments reported in this paper, in Appendix A.1 and A.2. Our algorithm is also outlined in Appendix A.2, and our implementations are available at: https://github.com/oozdenizci/PALL.

ACKNOWLEDGMENTS

This work was supported by the Graz Center for Machine Learning (GraML). This research was funded in whole or in part by the Austrian Science Fund (FWF) [10.55776/COE12].

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

# A APPENDIX

## A.1 EXPERIMENTAL SETUP

**Datasets:** We experimented with S-CIFAR10: 5 tasks $\times$ 2 classes, S-CIFAR100: 10 tasks $\times$ 10 classes (Krizhevsky, 2009), and S-TinyImageNet in various sequential configurations: 20 tasks $\times$ 10 classes, 40 tasks $\times$ 5 classes, and 100 tasks $\times$ 2 classes (Le & Yang, 2015). To adapt these datasets for our experimental setting, we first randomly allocate the classes in the dataset into disjoint tasks, and subsequently generate user request sequences $\mathcal{R}_{1:r}$ consisting of $T$ task learning and $N_u$ task unlearning instructions arranged in a logically consistent manner. This process was repeated in a randomized manner for each random seed (e.g., seed 0 to 19). We used standard data augmentation methods with random cropping and horizontal flipping of images.

**Model Architectures:** We used ResNet-18 and ResNet-34 architectures (He et al., 2016) in S-CIFAR10/100 experiments which are commonly used as benchmarks in lifelong learning, consisting of 11,164,352 and 21,311,168 trainable parameters respectively. In S-TinyImageNet experiments, we used a vision transformer (Dosovitskiy et al., 2020) based on the ViT-T/8 specifications from (Steiner et al., 2022), consisting of 12 encoder layers with 3 heads, a model width of 192, an MLP dimensionality of 768, and a patch size of 8x8, resulting in 5,408,064 trainable parameters. We did not have trainable parameters in the batch-norm or layer-norm layers of these models.

For architecture-based learning methods with subnetworks (i.e., PackNet, WSN, PALL), besides determining the connectivity of the linear layers within the ViT-T/8, we also used task-specific learnable class tokens and positional encoding layers altogether. This resulted in approximately 240,000 more parameters in $\boldsymbol{\theta}$ by default. None of the models were pretrained on other datasets.

## A.2 TRAINING CONFIGURATIONS AND IMPLEMENTATIONS

Our proposed methodology for privacy-aware lifelong learning is outlined in Algorithm 1.

**Model Training:** We use stochastic gradient descent (SGD) with momentum for optimization in S-CIFAR10/100 experiments, whereas an Adam optimizer was used in S-TinyImageNet experiments. We have found the use of an Adam optimizer with adaptive learning rates across the parameters and importance scores critical to facilitate the joint optimization for ViT-T/8. We perform training for 20 epochs per S-CIFAR10/100 task learning instruction with a batch size of 32, learning rate of 0.01, and weight decay with parameter 0.0005. We perform weight decay only on the non-frozen parameters. We did not perform regularization via weight decay on the importance scores. We perform training for 100 epochs per S-TinyImageNet task learning instruction with a batch size of 256, an initial learning rate of 0.001, and a cosine annealing learning rate scheduler.

**Baseline Methods:** We use the default elastic weight consolidation (EWC) (Kirkpatrick et al., 2017) lambda parameter of 100 for S-CIFAR10. For S-CIFAR100 with ResNet-34 and S-TinyImageNet with ViT-T/8, we set this parameter to 4000 and 1000 respectively, which yielded better results given the number of parameters in these models. We use the default learning without forgetting (LwF) (Li & Hoiem, 2017) regularization parameter of 1.0 and temperature parameter of 2.0. For learning with selective forgetting (LSF) (Shibata et al., 2021), we also use a combination of EWC and LwF for learning as performed in the original work, with the default LSF regularization weight of 10.

Baseline methods LwF and LSF both perform a preliminary classifier layer training phase at the beginning of task learning, which is then followed by complete model training. We also implemented these following the original methods, by splitting the number of epochs into two for the preliminary and subsequent phases, such that the total number of task learning epochs remains the same. Similarly for PackNet, we split the number of epochs into two since optimization involves two stages. After one round of training, magnitude-based pruning is performed, and then the model is finetuned (e.g., pretraining for 10 epochs, pruning, and finetuning for 10 epochs in S-CIFAR task learning).

**Unlearning Implementations:** For Sequential, EWC and LwF, we do not perform any changes to the model parameters for task unlearning. For EWC and LwF, we only discard the previously stored model weights (which are used for continual learning in these methods), corresponding to the unlearned task. For EWC, we also discard the previously stored Fisher information matrix corresponding to the unlearned task. We perform algorithm-specific unlearning updates for LSF, gradient

---

**Algorithm 1** Privacy-Aware Lifelong Learning (PALL)

---

1: **Input:** Datasets $\{\mathcal{D}^t\}_{t=1}^T$, neural network $f_{\boldsymbol{\theta}}$ with parameters $\boldsymbol{\theta} \in \mathbb{R}^d$, layer-wise subnetwork connectivity rate $\alpha$, retraining iterations $N_f$ and regularization weight $\beta$, parameter initialization distribution $\phi(.)$, total episodic memory buffer capacity $\sum_{t=1}^T |\mathcal{B}^t|$.

2: **Initialize:** $\Omega_0 = \emptyset$, $\mathcal{M}_0 = \emptyset$, $\mathbf{M}_0 = \mathbf{0}^d$, $\boldsymbol{\theta} \sim \phi(\boldsymbol{\theta})$

3: **for** $\mathcal{R}_i$ in $\mathcal{R}_{1:r} = [\mathcal{R}_1, \mathcal{R}_2, \ldots, \mathcal{R}_r]$ **do**

4:     **if** $\mathcal{R}_i = (t, \mathcal{D}^t, \mathbf{L})$ **then**                           # *Task Learning*

5:         $\Omega_i \leftarrow \Omega_{i-1} \cup \{t\}$

6:         Initialize parameter importance scores $\mathbf{s}_t \sim \phi(\mathbf{s}_t)$

7:         **for** $(\boldsymbol{x}, y) \sim \mathcal{D}^t$ **do**

8:             Obtain binary mask $\mathbf{m}_t(\mathbf{s}_t)$ with $\alpha$ connectivity rate per-layer using the largest $|\mathbf{s}_t|$

9:             Compute $\ell_{\text{ce}}(\boldsymbol{x}, y; \boldsymbol{\theta} \odot \mathbf{m}_t)$

10:            $\boldsymbol{\theta} \leftarrow \boldsymbol{\theta} - \eta \left( \frac{\partial \ell_{\text{ce}}}{\partial \boldsymbol{\theta}} \odot (1 - \mathbf{M}_{t-1}) \right)$

11:            $\mathbf{s}_t \leftarrow \mathbf{s}_t - \eta \left( \frac{\partial \ell_{\text{ce}}}{\partial \mathbf{s}_t} \right)$

12:         **end for**

13:         Obtain final binary mask $\mathbf{m}_t(\mathbf{s}_t)$ with $\alpha$ connectivity rate per-layer using the largest $|\mathbf{s}_t|$

14:         $\mathcal{M}_i \leftarrow \mathcal{M}_{i-1} \cup \{\mathbf{m}_t\}$

15:         $\mathbf{M}_i \leftarrow \mathbf{M}_{i-1} \vee \mathbf{m}_t$

16:         Store $\mathcal{B}^t = \{(\boldsymbol{x}_j, y_j, \boldsymbol{z}_j = f_{\boldsymbol{\theta} \odot \mathbf{m}_t}(\boldsymbol{x}_j)) \mid (\boldsymbol{x}_j, y_j) \sim \mathcal{D}^t\}_{1 \leq j \leq |\mathcal{B}^t|}$

17:         Reinitialize unused parameters $\boldsymbol{\theta} \odot (1 - \mathbf{M}_i)$ using $\phi(.)$

18:     **else if** $\mathcal{R}_i = (t, \mathbf{U})$ **then**                      # *Task Unlearning*

19:         $\Omega_i \leftarrow \Omega_{i-1} \setminus \{t\}$

20:         Delete episodic memory $\mathcal{B}^t$

21:         $\overline{\mathbf{m}}_t \leftarrow$ Retrieve the submask of $\mathbf{m}_t$ indicating the params trained via $\mathcal{D}^t$ or $\mathcal{B}^t$

22:         Reinitialize parameters $\boldsymbol{\theta} \odot \overline{\mathbf{m}}_t$ using $\phi(.)$

23:         $\bar{\boldsymbol{\theta}} \leftarrow$ Reinitialized params which were optimized on $\mathcal{D}^t$ and used by tasks $\tau > t, \tau \in \Omega_i$

24:         **for** $j = 1$ **to** $N_f$ **do**

25:             $\ell = \sum_{\substack{\tau > t \\ \tau \in \Omega_i}} \mathbb{E}_{(\boldsymbol{x}, y, \boldsymbol{z}) \sim \mathcal{B}^\tau} \left[ \ell_{\text{ce}} (\boldsymbol{x}, y; \boldsymbol{\theta} \odot \mathbf{m}_\tau) \right] + \beta \cdot \left[ \mathbb{E}_{(\boldsymbol{x}', y', \boldsymbol{z}') \sim \mathcal{B}^\tau} || f_{\boldsymbol{\theta} \odot \mathbf{m}_\tau}(\boldsymbol{x}') - \boldsymbol{z}' ||_2^2 \right]$,

26:             $\bar{\boldsymbol{\theta}} \leftarrow \bar{\boldsymbol{\theta}} - \eta \left( \frac{\partial \ell}{\partial \bar{\boldsymbol{\theta}}} \right)$

27:         **end for**

28:         $\mathcal{M}_i \leftarrow \mathcal{M}_{i-1} \setminus \{\mathbf{m}_t\}$

29:         $\mathbf{M}_i = \bigvee_{j \in \Omega_i} \mathbf{m}_j$

30:     **end if**

31: **end for**

---

episodic memory (GEM) (Lopez-Paz & Ranzato, 2017), experience replay (ER) (Chaudhry et al., 2019), dark experience replay (DER++) (Buzzega et al., 2020), when task unlearning is instructed. For task unlearning with GEM, ER and DER++, we implement an experience replay finetuning objective on the remaining tasks' episodic memories, and also predict uniform distributions using the unlearned task's episodic memories to accelerate forgetting. Task unlearning with LSF exploits its unique notion of manipulating the input space by superposing distinct auxiliary mnemonic codes on task-specific data. For unlearning, the mnemonic code of the unlearned task is directly provided to the network to be predicted incorrectly via a uniform label distribution, and the remaining task codes are directly provided to the network to be predicted accurately. We used a lower learning rate of 0.001 for these methods with S-CIFAR10/100, since it gave better results with unlearning mechanisms in place. All of these methods perform unlearning for $N_f$ iterations, similar to PALL.

During task unlearning with PALL, we use an episodic memory buffer with capacity of 500 and 1000 samples in S-CIFAR and S-TinyImageNet experiments, respectively. We partition the episodic memory buffer size a priori over all tasks, assuming that the amount of tasks $T$ is known beforehand. We perform a naive random sampling strategy to select the training set exemplars which will be stored in our task-specific episodic memory buffer (see Table A3 for alternatives). We used a Kaiming uniform distribution (He et al., 2015) for $\phi(.)$ while initializing the model parameters and importance scores at the beginning of training, as well as during unlearning-related parameter and

score resetting steps. Note that we keep the final classifier layer structure of the models hardwired based on the associated class neurons, such that there was no score optimization performed.

## A.3 Additional Experimental Results

We provide additional results based on our empirical evaluations. We investigate the impact of retraining after exact unlearning, hyperparameter configurations of the episodic memory rehearsal mechanism, an alternative exact unlearning baseline with model retraining from scratch using experience replay, scalability of PALL in longer lifelong learning and unlearning scenarios, training efficiency comparisons, and worst-case evaluation metrics.

### A.3.1 Impact of Retraining After Unlearning

In Table A1, we investigate the impact of the number of retraining iterations $N_f$, during rehearsal-based retraining of the reset parameters. We also define another evaluation metric in this context: $\mathcal{F}_u^{\max} = \max_{i \in \{2,\ldots,r\}, \mathcal{R}_i = (-,\mathbf{U})} \{\max_{t \in \Omega_i} (a_{i-1,t} - a_{i,t})\}$, which directly yields the maximum impact of any unlearning instruction on previously learned tasks. Results show that the absence of retraining after resetting the affected parameters ($N_f = 0$) does not perform well (i.e., $\mathcal{A}_l \uparrow$ / $\mathcal{F}_u^{\max} \downarrow$: 93.35/3.41 for S-CIFAR10, and 70.24/5.31 for S-CIFAR100). Therefore, a memory buffer and parameter retraining is necessary to balance exact unlearning. We could also achieve better performances after unlearning by using longer retraining durations (e.g., $\mathcal{A}_l \uparrow$ / $\mathcal{F}_u^{\max} \downarrow$ for S-CIFAR100 with $N_f = 100$: 72.46/2.39, as opposed to our default choice $N_f = 50$: 72.35/2.17).

Table A1: Experiments with PALL ($\alpha = 1/T$) on the impact of $N_f$ for parameter retraining after unlearning.

| | **S-CIFAR10** ($T = 5$) | | | **S-CIFAR100** ($T = 10$) | | |
|---|---|---|---|---|---|---|
| | $\mathcal{A}_l \uparrow$ | $\mathcal{F}_u \downarrow$ | $\mathcal{F}_u^{\max} \downarrow$ | $\mathcal{A}_l \uparrow$ | $\mathcal{F}_u \downarrow$ | $\mathcal{F}_u^{\max} \downarrow$ |
| $N_f = 0$ | 93.35 | 1.07 | 3.41 | 70.24 | 1.44 | 5.31 |
| $N_f = 10$ | 93.84 | 0.72 | 2.26 | 71.65 | 0.82 | 3.85 |
| $N_f = 25$ | 93.96 | 0.72 | 2.30 | 72.04 | 0.58 | 2.57 |
| $N_f = 50$ | 94.34 | 0.60 | 1.94 | 72.35 | 0.40 | 2.17 |
| $N_f = 100$ | 94.15 | 0.56 | 1.68 | 72.46 | 0.38 | 2.39 |

**On the Retrained Parameters After Unlearning:** In Table A2 we provide details on the ratio of the affected parameters $\bar{\boldsymbol{\theta}}$ during unlearning, which required retraining using memory rehearsal. Specifically, we observed that to facilitate task unlearning, PALL only requires retraining of 4.07%, 1.98% and 1.03% of the overall model parameters on average, in S-CIFAR10/100 and S-TinyImageNet experiments respectively. This indicates a very efficient retraining phase with sparse gradient updates for $N_f$ iterations, leading to a beneficial increase in performance (from Table A1, S-CIFAR10 with $N_f = 0$: 93.35 vs $N_f = 50$: 94.34, S-CIFAR100 with $N_f = 0$: 70.24 vs $N_f = 50$: 72.35). We also investigated the values of these parameters before and after retraining, to verify that the parameters do not converge back to their pre-unlearning values, which would conflict with unlearning. We observed that the mean absolute difference values were non-zero overall, indicating a feasible retraining mechanism to complement our unlearning approach.

Table A2: Ratio of retrained params on average, and the mean absolute difference between these params before and after retraining. Results are for PALL ($\alpha = 1/T$), averaged over $N_u = 3$ requests for 10 seeds.

| | **S-CIFAR10** w/ResNet-18 | **S-CIFAR100** w/ResNet-34 | **S-TinyImageNet** w/ViT-T/8 |
|---|---|---|---|
| # total params: $d$ | 11.2M | 21.3M | 5.4M |
| # subnetwork params: $d/T$ | 2.24M (20%) | 2.13M (10%) | 270K (5%) |
| # retrained params: $|\bar{\boldsymbol{\theta}}|$ | ~454.4K | ~421.8K | ~55.5K |
| ratio of retrained params | 4.07% | 1.98% | 1.03% |
| mean absolute difference ($\uparrow$) | 0.0019 | 0.0021 | 0.0214 |

### A.3.2 Ablations on the Episodic Memory Rehearsal Method

In Table A3 we investigate the influence of the random exemplar sampling method used to select the data samples to be stored in the episodic memory buffer, in comparison to four methods that use different criteria: (1) *Herding algorithm* from iCaRL (Rebuffi et al., 2017), which sorts the samples from a class to select more representative exemplars based on the distance to the mean sample of that class, using feature representations of the penultimate layer. (2) *Entropy-based* selection using

Table A4: Ablation experiments on the memory rehearsal based retraining stage. We present evaluations on PALL ($\alpha = 1/T$) with $N_u = 3$. Results are averaged over 20 random seeds.

| Episodic Memory $\sum_{t=1}^{T} |\mathcal{B}^t|$ | Method | Exact Unlearning | S-CIFAR10 $\mathcal{A}_l \uparrow$ | $\mathcal{F}_l \downarrow$ | $\mathcal{F}_u \downarrow$ | S-CIFAR100 $\mathcal{A}_l \uparrow$ | $\mathcal{F}_l \downarrow$ | $\mathcal{F}_u \downarrow$ |
|---|---|---|---|---|---|---|---|---|
| – | Sequential | ✗ | 70.71 | 13.86 | 0.0 | 35.35 | 13.43 | 0.0 |
| 200 | ER | ✗ | 84.09 | 4.83 | 2.56 | 50.08 | 6.69 | 8.14 |
| | DER++ | ✗ | 89.23 | 3.09 | 1.03 | 59.82 | 6.99 | 1.43 |
| | **PALL** ($\beta = 0.0$) | ✓ | 93.91 | 0.0 | 2.36 | 71.68 | 0.0 | 0.83 |
| | **PALL** ($\beta = 0.5$) | ✓ | 93.95 | 0.0 | 0.67 | 71.90 | 0.0 | 0.72 |
| | **PALL** ($\beta = 1.0$) | ✓ | 93.36 | 0.0 | 3.35 | 71.73 | 0.0 | 0.79 |
| 500 | ER | ✗ | 87.88 | 3.45 | 1.61 | 57.63 | 4.67 | 7.91 |
| | DER++ | ✗ | 92.04 | 1.62 | 0.66 | 66.84 | 4.52 | 0.95 |
| | **PALL** ($\beta = 0.0$) | ✓ | 94.12 | 0.0 | 1.98 | 71.98 | 0.0 | 0.50 |
| | **PALL** ($\beta = 0.5$) | ✓ | 94.34 | 0.0 | 0.60 | 72.35 | 0.0 | 0.40 |
| | **PALL** ($\beta = 1.0$) | ✓ | 93.60 | 0.0 | 2.86 | 72.32 | 0.0 | 0.50 |
| 1000 | ER | ✗ | 89.76 | 2.52 | 1.22 | 63.10 | 3.31 | 7.18 |
| | DER++ | ✗ | 93.20 | 1.00 | 0.63 | 70.96 | 3.01 | 0.89 |
| | **PALL** ($\beta = 0.0$) | ✓ | 94.36 | 0.0 | 1.37 | 72.50 | 0.0 | 0.29 |
| | **PALL** ($\beta = 0.5$) | ✓ | 94.25 | 0.0 | 0.54 | 72.58 | 0.0 | 0.23 |
| | **PALL** ($\beta = 1.0$) | ✓ | 93.62 | 0.0 | 2.91 | 72.52 | 0.0 | 0.27 |
| – | Independent | ✓ | 95.19 | 0.0 | 0.0 | 73.22 | 0.0 | 0.0 |

the output logits (Chaudhry et al., 2018), which selects exemplars with higher output distribution uncertainty. (3) *Confidence-based* selection (Wang et al., 2022), which selects exemplars with higher output logit confidence. (4) *Logit distance* based selection (Chaudhry et al., 2018), which selects critical exemplars closer to the decision boundary. Overall, our results in Table A3 did not indicate significant improvements in performance after incorporating such prioritized sampling mechanisms while accumulating the episodic memory buffer.

In Table A4 we present our ablation experiments on the quantitative hyperparameters of the memory rehearsal mechanism, based on S-CIFAR10/100 simulations.

Table A3: Influence of the buffer sampling strategy on PALL ($\alpha = 1/T$), based on the default hyperparameters. Results are averaged over 10 random seeds for these comparisons.

| | S-CIFAR10 $\mathcal{A}_l \uparrow$ | $\mathcal{F}_u \downarrow$ | $\mathcal{F}_u^{\max} \downarrow$ | S-CIFAR100 $\mathcal{A}_l \uparrow$ | $\mathcal{F}_u \downarrow$ | $\mathcal{F}_u^{\max} \downarrow$ |
|---|---|---|---|---|---|---|
| Random selection | 94.25 | 0.57 | 1.80 | 72.11 | 0.49 | 2.32 |
| Herding algorithm | 94.15 | 0.50 | 1.74 | 72.10 | 0.70 | 3.14 |
| Entropy-based | 85.15 | 4.71 | 15.32 | 70.55 | 1.33 | 4.22 |
| Confidence-based | 83.54 | 7.28 | 27.63 | 70.21 | 1.46 | 5.70 |
| Logit distance | 81.83 | 7.17 | 24.34 | 70.01 | 1.69 | 6.20 |

Here we consider the rehearsal buffer samples to be selected via random sampling. Our experimental results in the main text were based on $\beta = 0.5$ and a buffer capacity of 500 for these datasets. Our ablation experiment results in Table A4 also support that $\beta = 0.5$, i.e., experience memory rehearsal via the DER++ method (Buzzega et al., 2020), performs consistently better than the other alternatives in all cases.

Notably, the results with $\beta = 0.0$, i.e., vanilla experience replay (Chaudhry et al., 2019), demonstrates the quantitative evaluations for the case where we do not store any episodic logit values in the buffer, but only use $(\boldsymbol{x}, y)$ pairs. These results were also found relatively good, in case the episodic memory buffer does not allow storing logits (Chaudhry et al., 2019) (e.g., with S-CIFAR10, $\mathcal{A}_l \uparrow$ metrics for buffer size 1000, $\beta = 0.0$: 93.91, $\beta = 0.5$: 93.95, $\beta = 1.0$: 93.36).

**Retraining with Experience Replay for Exact Unlearning:** We investigate an alternative approach to adapt the traditional experience replay (ER) (Chaudhry et al., 2019) method upon task unlearning requests. In our main experiments, ER was adapted to perform *approximate* unlearning, by finetuning model parameters using episodic memories towards misclassification on the task to be unlearned. In this new setting, for any unlearning request, we retrain the entire model from scratch using the experience replay buffer. We test this as an alternative baseline that can satisfy exact unlearning,

Table A6: Comparisons of PALL with baseline methods on S-TinyImageNet (100 tasks × 2 classes) for various number of unlearning instructions in a longer lifelong learning scenario. Results are averaged over 10 random seeds for these comparisons.

| | Exact Unlearning | $N_u = 3$ | | | $N_u = 5$ | | | $N_u = 10$ | | | $N_u = 20$ | | | $N_u = 50$ | | |
|---|---|---|---|---|---|---|---|---|---|---|---|---|---|---|---|---|
| | | $\mathcal{A}_l$ | $\mathcal{F}_l$ | $\mathcal{F}_u$ | $\mathcal{A}_l$ | $\mathcal{F}_l$ | $\mathcal{F}_u$ | $\mathcal{A}_l$ | $\mathcal{F}_l$ | $\mathcal{F}_u$ | $\mathcal{A}_l$ | $\mathcal{F}_l$ | $\mathcal{F}_u$ | $\mathcal{A}_l$ | $\mathcal{F}_l$ | $\mathcal{F}_u$ |
| Sequential | ✗ | 52.46 | 1.71 | 0.0 | 51.86 | 1.72 | 0.0 | 53.01 | 1.73 | 0.0 | 54.15 | 1.73 | 0.0 | 51.91 | 1.80 | 0.0 |
| EWC | ✗ | 79.53 | 0.43 | 0.0 | 79.57 | 0.43 | 0.0 | 79.53 | 0.43 | 0.0 | 79.59 | 0.43 | 0.0 | 79.17 | 0.44 | 0.0 |
| DER++ | ✗ | 76.58 | 0.64 | 0.45 | 75.04 | 0.62 | 0.16 | 75.63 | 0.64 | 0.36 | 74.95 | 0.57 | 0.45 | 73.22 | 0.54 | 0.22 |
| PackNet | ✗ | 81.73 | 0.0 | 0.0 | 81.76 | 0.0 | 0.0 | 81.85 | 0.0 | 0.0 | 81.97 | 0.0 | 0.0 | 81.75 | 0.0 | 0.0 |
| WSN | ✗ | 88.05 | 0.0 | 0.0 | 87.97 | 0.0 | 0.0 | 87.62 | 0.0 | 0.0 | 88.02 | 0.0 | 0.0 | 87.74 | 0.0 | 0.0 |
| **PALL** ($\alpha = 0.01$) | ✓ | 85.89 | 0.0 | 0.53 | 85.45 | 0.0 | 0.41 | 85.18 | 0.0 | 0.27 | 84.21 | 0.0 | 0.21 | 81.36 | 0.0 | 0.17 |
| **PALL** ($\alpha = 0.025$) | ✓ | 86.11 | 0.0 | 0.43 | 85.84 | 0.0 | 0.31 | 85.52 | 0.0 | 0.22 | 84.29 | 0.0 | 0.19 | 79.77 | 0.0 | 0.20 |
| **PALL** ($\alpha = 0.05$) | ✓ | 85.80 | 0.0 | 0.35 | 85.60 | 0.0 | 0.29 | 85.04 | 0.0 | 0.23 | 84.00 | 0.0 | 0.20 | 77.96 | 0.0 | 0.24 |
| Independent | ✓ | 86.80 | 0.0 | 0.0 | 86.51 | 0.0 | 0.0 | 86.53 | 0.0 | 0.0 | 86.71 | 0.0 | 0.0 | 86.93 | 0.0 | 0.0 |

Header above table: **S-TinyImageNet** ($T = 100$)

due to resetting the entire model upon any unlearning request, which is analogous to the *model retraining* baselines from the machine unlearning literature (Bourtoule et al., 2021). Our results in Table A5 show that this approach, annotated as "ER (Retraining)", does not scale well even for the slightly more complex S-CIFAR100 task: $\mathcal{A}_l = 34.50$. This is due to the model being repeatedly reset, and thus the memory buffer becoming the only dataset for the learner.

Table A5: Evaluating *model retraining with experience replay*, i.e., ER (Retraining), in comparison to Independent model training, PALL, and our original implementation of ER with approximate unlearning.

| | **S-CIFAR10** ($T = 5$) | | | | **S-CIFAR100** ($T = 10$) | | | |
|---|---|---|---|---|---|---|---|---|
| | $\mathcal{A}_l \uparrow$ | $\mathcal{A}_u \downarrow$ | $\mathcal{F}_l \downarrow$ | $\mathcal{F}_u \downarrow$ | $\mathcal{A}_l \uparrow$ | $\mathcal{A}_u \downarrow$ | $\mathcal{F}_l \downarrow$ | $\mathcal{F}_u \downarrow$ |
| Independent | 95.19 | *Exact* | **0.0** | 0.0 | 73.22 | *Exact* | **0.0** | 0.0 |
| **PALL** ($\alpha = 1/T$) | 94.34 | *Exact* | **0.0** | 0.60 | 72.35 | *Exact* | **0.0** | 0.40 |
| ER (Retraining) | 78.80 | *Exact* | 2.05 | 7.53 | 34.50 | *Exact* | 4.07 | 20.40 |
| ER | 87.88 | 58.48 | 3.45 | 1.61 | 57.63 | 42.69 | 4.67 | 7.91 |

### A.3.3 Scalability to Longer Lifelong Learning and Unlearning Scenarios

We investigate the scalability of PALL on longer lifelong learning and unlearning scenarios by testing it on the S-TinyImageNet dataset with 100 tasks. Our results are presented in Table A6 with increasing number of task unlearning requests. We overall show that there is no limitation in the scalability of PALL, and the algorithm performs well in learning to sub-partition tasks within one ViT-T/8 architecture. We observed that the best performing PALL model is often the one with 97.5% sparse task-specific subnetworks ($\alpha = 0.025$) with knowledge transfer, and the $\mathcal{A}_l$ metric occasionally degrades as the number of unlearning requests increase in the sequence, due to the parameter retraining process. In the extreme experimental condition, for a user request sequence with 100 task learning and $N_u = 50$ unlearning requests, PALL achieved an observed performance of 81.36/0.17 for $\mathcal{A}_l \uparrow$ / $\mathcal{F}_u \downarrow$, with $\alpha = 0.01$ (i.e., maximum of 54K params per task).

Our approach also performs favorably in comparison to the baselines in such a longer scenario, when all evaluation criteria are considered together (no catastrophic forgetting, forward knowledge transfer, exact unlearning and memory-efficiency). Specifically in Table A6, although PALL yields lower $\mathcal{A}_l$ than WSN as the number of unlearning requests increase (e.g., $N_u = 20$, WSN: 88.02, PALL ($\alpha = 0.025$): 84.29), our approach can provide exact unlearning guarantees by design. On the other hand, although training Independent models can also ensure exact unlearning, it becomes a significantly more memory-demanding approach in this longer scenario. Specifically, the base model has an inference time model size of 20.63 MB, which corresponds to 32-bit representation of its approximately 5.4M parameters. As the number of tasks increase, the model size increment for PALL was observed to be approximately 0.69 MB for each additional task (due to the binary mask parameter size increment), which would result in a maximum total model size of 89.53 MB at the end of the sequence with $T = 100$. On the other hand, training independent models with $d \times T$ parameters (32-bit) would result in a model size of 2063 MB, with a step-wise increase of 20.63 MB for each new task. Thus, we argue that our approach does scale well in longer lifelong learning and unlearning scenarios by offering a reasonable trade-off in overall performance.

Table A7: Comparisons of training times while processing a single task learning or unlearning request. We use ResNet-18 with S-CIFAR10, ResNet-34 with S-CIFAR100, and ViT-T/8 with S-TinyImageNet. We use PALL with $\alpha = 1/T$, and $N_f = 50$ retraining iterations after unlearning.

| | **S-CIFAR10** $(T = 5)$ | | **S-CIFAR100** $(T = 10)$ | | **S-TinyImageNet** $(T = 20)$ | |
| | Learning $\mathcal{R}_i = (-, \mathbf{L})$ | Unlearning $\mathcal{R}_i = (-, \mathbf{U})$ | Learning $\mathcal{R}_i = (-, \mathbf{L})$ | Unlearning $\mathcal{R}_i = (-, \mathbf{U})$ | Learning $\mathcal{R}_i = (-, \mathbf{L})$ | Unlearning $\mathcal{R}_i = (-, \mathbf{U})$ |
|---|---|---|---|---|---|---|
| Sequential | 3.23 sec/epoch | $< 10^{-4}$ sec | 2.91 sec/epoch | $< 10^{-4}$ sec | 2.71 sec/epoch | $< 10^{-4}$ sec |
| EWC | 6.42 sec/epoch | $< 10^{-4}$ sec | 5.59 sec/epoch | $< 10^{-4}$ sec | 3.59 sec/epoch | $< 10^{-4}$ sec |
| LwF | 4.87 sec/epoch | $< 10^{-4}$ sec | 4.23 sec/epoch | $< 10^{-4}$ sec | 3.37 sec/epoch | $< 10^{-4}$ sec |
| LSF | 9.02 sec/epoch | 0.51 sec | 8.21 sec/epoch | 0.90 sec | 5.28 sec/epoch | 1.84 sec |
| GEM | 23.8 sec/epoch | 0.93 sec | 12.4 sec/epoch | 1.64 sec | 3.19 sec/epoch | 2.10 sec |
| ER | 6.65 sec/epoch | 0.94 sec | 5.70 sec/epoch | 1.73 sec | 2.95 sec/epoch | 2.13 sec |
| DER++ | 9.09 sec/epoch | 1.38 sec | 8.44 sec/epoch | 2.44 sec | 3.11 sec/epoch | 3.06 sec |
| PackNet | 3.40 sec/epoch | $< 10^{-4}$ sec | 3.07 sec/epoch | $< 10^{-4}$ sec | 2.65 sec/epoch | $< 10^{-4}$ sec |
| WSN | 32.6 sec/epoch | $< 10^{-4}$ sec | 30.2 sec/epoch | $< 10^{-4}$ sec | 2.96 sec/epoch | $< 10^{-4}$ sec |
| **PALL** | 33.1 sec/epoch | 1.09 sec | 30.8 sec/epoch | 1.88 sec | 3.13 sec/epoch | 2.59 sec |
| Independent | 3.20 sec/epoch | $< 10^{-4}$ sec | 2.89 sec/epoch | $< 10^{-4}$ sec | 2.66 sec/epoch | $< 10^{-4}$ sec |

### A.3.4 COMPARISONS OF TRAINING TIMES

We compare training times of each algorithm in Table A7 using an NVIDIA A40 GPU. We evaluate all algorithms on the same test sequence that begins as: $\mathcal{R}_{1:r} = [(1, \mathcal{D}^1, \mathbf{L}), (2, \mathcal{D}^2, \mathbf{L}), (1, \mathbf{U}), \ldots]$. We then compute the per-epoch training duration during processing of the second task learning request $(2, \mathcal{D}^2, \mathbf{L})$, and the total time elapsed during processing of the task unlearning request $(1, \mathbf{U})$. Since these timings depend on the model architecture, we present them for all datasets.

During task learning, PALL incurs an additional training time overhead to optimize task-specific subnetworks with knowledge transfer. This is also observed in WSN (Kang et al., 2022) (e.g., for S-TinyImageNet, WSN: 2.96 sec/epoch, PALL: 3.13 sec/epoch). Both algorithms have a significantly higher training time than the other baselines due to the parameter-level gradient masking within each gradient update step. This time difference is larger in ResNet architectures with convolutional layer masking, as opposed to the ViT-T/8 architectures, where only dense layers throughout the model are being masked.

During unlearning, the total retraining time of PALL varies between 1 to 3 sec when $N_f = 50$ is used, which is a negligible difference to other methods.

We performed further comparisons of the change in training times as the number of tasks increase, in Figure A1. We conducted these simulations on S-TinyImageNet ($T = 100$) based on the models from Table A6. We did not include DER++, since the training time of this algorithm was too far out of the range from the other methods, and also rapidly increased with the number of tasks (e.g., 3.04 sec/epoch when 10 tasks are learned, 6.19 sec/epoch for 30 tasks are learned, etc.).

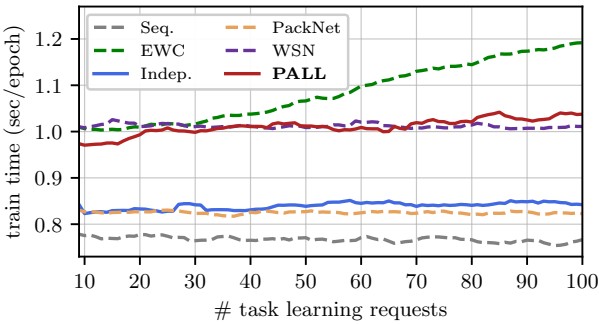

Figure A1: Comparisons of task learning durations on S-TinyImageNet ($T = 100$), as the number of tasks increase. Methods indicated with solid lines (Indep. & PALL) can ensure exact unlearning upon request.

Overall, we observe that PALL maintains a relatively stable (although minimally fluctuating) training time overhead caused by the parameter gradient masking. This masking operation is implemented based on the cumulative binary mask matrix $\mathbf{M}_i$, which is sequentially updated during each task learning request in a single step via: $\mathbf{M}_i = \bigvee_{j \in \Omega_{i-1}} \mathbf{m}_j$, and then applied to the gradients. Therefore, since this series of logical operations do not influence the execution time of the overall algorithm drastically, the impact of the increase in the number of tasks is negligible for PALL.

### A.3.5 WORST-CASE EVALUATION METRICS ACROSS DIFFERENT REQUEST SEQUENCES

In Table A8, we present the worst-case evaluation metrics corresponding to our results in Table 1 of the main text. We specifically present the minimum task learning performance metric observed across randomized seed repetitions ($\mathcal{A}_l^{\min}$), and the worst-case impact of task unlearning ($\mathcal{F}_u^{\max}$) as previously defined in Appendix A.3.1. These metrics specifically correspond to certain user request sequences of interleaved learning and unlearning instructions, where the models fail to perform the most. Since PALL consistently yields no catastrophic forgetting ($\mathcal{F}_l = 0$) and unlearning is exact (i.e., $\mathcal{A}_u$ is irrelevant), we do not include the other metrics.

In Table A8, we observe that the worst-case evaluations are mainly consistent with the averaged metrics

Table A8: Worst-case evaluation metrics for task learning performance ($\mathcal{A}_l^{\min}$), and impact of task unlearning ($\mathcal{F}_u^{\max}$), observed across randomized repetitions of user sequences. We consider PALL with $\alpha = 1/T$.

| | S-CIFAR10 (T = 5) | | S-CIFAR100 (T = 10) | | S-TinyImageNet (T = 20) | |
|---|---|---|---|---|---|---|
| | $\mathcal{A}_l^{\min} \uparrow$ | $\mathcal{F}_u^{\max} \downarrow$ | $\mathcal{A}_l^{\min} \uparrow$ | $\mathcal{F}_u^{\max} \downarrow$ | $\mathcal{A}_l^{\min} \uparrow$ | $\mathcal{F}_u^{\max} \downarrow$ |
| Sequential | 36.78 | 0.0 | 21.03 | 0.0 | 13.93 | 0.0 |
| EWC | 28.60 | 0.0 | 51.79 | 0.0 | 52.18 | 0.0 |
| LwF | 83.43 | 0.0 | 48.83 | 0.0 | 41.04 | 0.0 |
| LSF | 81.50 | 3.13 | 51.53 | 8.87 | 35.42 | 11.60 |
| GEM | 79.85 | 5.71 | 52.10 | 10.28 | 39.52 | 12.94 |
| ER | 80.63 | 5.23 | 49.87 | 15.07 | 39.79 | 12.12 |
| DER++ | 84.45 | 2.22 | 61.57 | 4.58 | 44.46 | 7.11 |
| PackNet | 89.85 | 0.0 | 72.99 | 0.0 | 58.35 | 0.0 |
| WSN | 86.75 | 0.0 | 69.93 | 0.0 | 61.34 | 0.0 |
| **PALL** | 89.95 | 1.94 | 69.83 | 2.17 | 59.91 | 4.42 |
| Independent | 90.85 | 0.0 | 69.46 | 0.0 | 60.29 | 0.0 |

presented in the Table 1 of the main text. For instance, for S-CIFAR10, PALL yields the best worst-case $\mathcal{A}_l^{\min}$ metric with 89.95 among all methods, close to the upper bound baseline performance with Independent: 90.85. Similarly for S-CIFAR100, PALL achieves an $\mathcal{A}_l^{\min}$ of 69.83, whereas Independent: 69.46. In terms of the $\mathcal{F}_u^{\max}$ metric, we observe that PALL is consistently better on all datasets, demonstrating the robustness of the retraining step following unlearning.

