# OpenReview forum: "Privacy-Aware Lifelong Learning"
_ICLR.cc/2025/Conference — ICLR 2025 Poster_

### Official Review · Reviewer_iYfp · 2024-10-25

**Soundness:** 3
**Presentation:** 3
**Contribution:** 3
**Rating:** 8
**Confidence:** 4

**Summary:**

This manuscript focused on a novel setting for task-incremental lifelong learning with selectively forgetting (i.e., exact task unlearning) regarding the data privacy concern. This setting aims to simultaneously realize some challenging objectives, including learning tasks without forgetting, facilitating forward knowledge transfer, selective forgetting of specific tasks, and guaranteeing memory efficiency. To address this challenging task, the authors proposed Privacy-Aware Lifelong Learning (PALL) framework. Specifically, inspired by previous work WSN, this method adopted an architecture-based strategy through knowledge isolation and sharing with subnetwork masks. Furthermore, the experience replay strategy was applied to mitigate potential performance degradation caused by the parameter reinitialization after knowledge unlearning. Empirical experiments on three standard benchmarks were conducted to support the effectiveness in different aspects compared to other baseline methods.

**Strengths:**

1. This manuscript considered a novel setting. The data privacy concern is practical in lifelong learning but the exact task unlearning is less explored in this context. The proposed setting is well-defined.

2. The adoption of subnetwork masks provided a unified perspective for controlling catastrophic forgetting and exact task unlearning.

3. The experiments indicated that the proposed method is memory-efficient compared to baseline methods.

**Weaknesses:**

1. Most parts of this manuscript came from a previous study WSN [1]. Although the experience replay (in section 3.3) was tailored for knowledge recovery after an unlearning step, the new message conveyed from this manuscript is limited.

2. The scalability of the proposed method needs to be further demonstrated with more experiments on other datasets.

3. The current version only considered the task-incremental setting where the task id is known during the inference. I wonder the applicability on other settings, such as class-incremental setting or domain incremental setting.

**Questions:**

1. I noticed that the maximum number of tasks in this manuscript is 40 in S-TinyImage and the maximum unlearning requests $N_u$ is 9 for the S-TinyImageNet 20-split setting. I wonder if the proposed method is scalable for larger numbers of tasks and more unlearning requests, e.g., on the Omniglot-Rotation dataset which has 100 tasks in total.

2. Can the proposed PALL be adapted to the class-incremental setting? If so, what changes should be introduced to make it compatible with the scenario if we don't know the task identity?

3. About the training efficiency. In section 5, the authors only discussed the memory that needs to store the additional parameters along the learning dynamic. The comparisons of the training time among different methods were missed. The proposed method is mainly based on WSN [1]. However, according to my experience in training WSN, I found that WSN could be a little slow since it requires the parameter-level gradient masking within each backpropagation step and I believe this operation also exists in your proposed PALL. Thus, could the authors provide a comparison w.r.t. the training efficiency among different methods?

References:

[1] Forget-free Continual Learning with Winning Subnetworks. ICML 2023.

---

> ### Author Response · Authors · 2024-11-19
> **Response to Reviewer iYfp (part 1/2)**
>
> >_**(Q1)** I noticed that the maximum number of tasks in this manuscript is 40 in S-TinyImage and the maximum unlearning requests $N_u$ is 9 for the S-TinyImageNet 20-split setting. I wonder if the proposed method is scalable for larger numbers of tasks and more unlearning requests, e.g., on the Omniglot-Rotation dataset which has 100 tasks in total._
>
> >_**(W2)** The scalability of the proposed method needs to be further demonstrated with more experiments on other datasets._
>
> We now performed additional experiments with S-TinyImageNet (100 tasks) using ViT-T/8, also for increased number of unlearning requests $N_u$. Results are provided in the table below, for the metrics $\mathcal{A}_l / \mathcal{F}_u$ in each experimental setting. We could demonstrate that there is no particular limitation of implementing PALL in larger scale settings. In the extreme experimental setting, for a user request sequence with 100 task learning and $N_u=50$ task unlearning requests, PALL ($\alpha=0.01$) achieves $\mathcal{A}_l=81.36\%$ by using at most 54K parameters per task.
>
> We included these results as **Table A6** in the new Appendix A.3.3. We also partially discussed these results under Section 5.3 of the main text.
>
> |  S-TinyImageNet ($T=100$) |   $N_u=3$  |   $N_u=5$  |  $N_u=10$  |  $N_u=20$  |  $N_u=50$  |
> |:-------------------------:|:----------:|:----------:|:----------:|:----------:|:----------:|
> |  **PALL** ($\alpha=0.01$) | 85.89/0.53 | 85.45/0.41 | 85.18/0.27 | 84.21/0.21 | 81.36/0.17 |
> | **PALL** ($\alpha=0.025$) | 86.11/0.43 | 85.84/0.31 | 85.52/0.22 | 84.29/0.19 | 79.77/0.20 |
> |  **PALL** ($\alpha=0.05$) | 85.80/0.35 | 85.60/0.29 | 85.04/0.23 | 84.00/0.20 | 77.96/0.24 |
>
> >_**(Q2)** Can the proposed PALL be adapted to the class-incremental setting? If so, what changes should be introduced to make it compatible with the scenario if we don't know the task identity?_
>
> >_**(W3)** The current version only considered the task-incremental setting where the task id is known during the inference. I wonder the applicability on other settings, such as class-incremental setting or domain incremental setting._
>
> We now discuss a future direction for PALL to be applicable in class- and domain-incremental learning settings in the last paragraph of the Discussion section of our paper as follows:
>
> *''In this work, we focus on task-incremental learning settings, where the task ID is available to the learner.
> Our approach is not yet readily applicable to a class- or domain-incremental scenario where all previously learned tasks' label spaces are unified, since PALL disentangles the choice of the task-specific subnetwork based on the task IDs to ensure exact unlearning.
> To be applicable in these settings, PALL can be extended with a privacy-aware auxiliary algorithm to first identify the task, and subsequently utilize task-specific subnetwork via gating (Aljundi et al., 2017; Von Oswald et al., 2019).''*
>
> Aljundi et al., ''Expert Gate: Lifelong Learning with a Network of Experts'', CVPR 2017.
>
> Von Oswald et al., ''Continual learning with hypernetworks'', ICLR 2020.

---

> ### Author Response · Authors · 2024-11-19
> **Response to Reviewer iYfp (part 2/2)**
>
> >_**(Q3)** About the training efficiency. In section 5, the authors only discussed the memory that needs to store the additional parameters along the learning dynamic. The comparisons of the training time among different methods were missed. The proposed method is mainly based on WSN [1]. However, according to my experience in training WSN, I found that WSN could be a little slow since it requires the parameter-level gradient masking within each backpropagation step and I believe this operation also exists in your proposed PALL. Thus, could the authors provide a comparison w.r.t. the training efficiency among different methods?_
>
> We now added training time comparisons and discussions in the new subsection Appendix A.3.4 with **Table A8**. This table is also provided below, where we indicate the "task learning time per epoch / total task unlearning time" for each dataset. For total task unlearning times which were less than $10^{-4}$ seconds, we only write "$-$".
>
> Indeed, during task learning, PALL incurs an additional training time overhead to optimize task-specific subnetworks with knowledge transfer. This is also observed in WSN (e.g., for S-TinyImageNet, WSN: 2.96 sec/epoch, PALL: 3.13 sec/epoch). Both algorithms have a higher training time than the other baselines, since there is a parameter-level gradient masking within each gradient update step. This difference is larger in ResNets with convolutional layer masking, as opposed to the ViT-T/8 where only dense layers throughout the model are being masked. During unlearning, the total retraining time of PALL has a negligible difference to other methods.
>
> |             | S-CIFAR10 ($T=5$)         | S-CIFAR100 ($T=10$)       | S-TinyImageNet ($T=20$)   |
> |-------------|---------------------------|---------------------------|---------------------------|
> | Sequential  | 3.23 sec/epoch / $-$      | 2.91 sec/epoch / $-$      | 2.71 sec/epoch / $-$      |
> | EWC         | 6.42 sec/epoch / $-$      | 5.59 sec/epoch / $-$      | 3.59 sec/epoch / $-$      |
> | LwF         | 4.87 sec/epoch / $-$      | 4.23 sec/epoch / $-$      | 3.37 sec/epoch / $-$      |
> | LSF         | 9.02 sec/epoch / 0.51 sec | 8.21 sec/epoch / 0.90 sec | 5.28 sec/epoch / 1.84 sec |
> | GEM         | 23.8 sec/epoch / 0.93 sec | 12.4 sec/epoch / 1.64 sec | 3.19 sec/epoch / 2.10 sec |
> | ER          | 6.65 sec/epoch / 0.94 sec | 5.70 sec/epoch / 1.73 sec | 2.95 sec/epoch / 2.13 sec |
> | DER++       | 9.09 sec/epoch / 1.38 sec | 8.44 sec/epoch / 2.44 sec | 3.11 sec/epoch / 3.06 sec |
> | PackNet     | 3.40 sec/epoch / $-$      | 3.07 sec/epoch / $-$      | 2.65 sec/epoch / $-$      |
> | WSN         | 32.6 sec/epoch / $-$      | 30.2 sec/epoch / $-$      | 2.96 sec/epoch / $-$      |
> | **PALL ($\alpha=1/T$)**    | 33.1 sec/epoch / 1.09 sec | 30.8 sec/epoch / 1.88 sec | 3.13 sec/epoch / 2.59 sec |
> | Independent | 3.20 sec/epoch / $-$      | 2.89 sec/epoch / $-$      | 2.66 sec/epoch / $-$      |
>
>
> >_**(W1)** Most parts of this manuscript came from a previous study WSN [1]. Although the experience replay (in section 3.3) was tailored for knowledge recovery after an unlearning step, the new message conveyed from this manuscript is limited._
>
> We introduce a novel problem configuration (i.e., task-incremental lifelong learning with exact unlearning), and present a new solution in this setting. We adapt importance score optimization based sparse structure learning algorithms to solve this problem (Ramanujan et al. 2020), which were also explored in traditional lifelong learning by (Kang et al., 2022). We acknowledged both of these works in our submission, and clearly discussed the distinction and innovation of our work in the Discussion section as follows:
>
> *''Our method is partially based on architecture-based lifelong learning methods that allow selective knowledge transfer (Kang et al., 2022; Ramanujan et al. 2020), which we innovatively extended into a hybrid learning strategy that is also equipped with a rehearsal-based lifelong learning method (Buzzega et al., 2020). This enables our algorithm to handle exact task unlearning requests in the presence of knowledge transfer, which was not addressed to date. Additionally, we also performed critical modifications to the existing subnetwork optimization methods, such as reinitializing the scores and unused weights after each learning request. These were required to satisfy overall exact unlearning guarantees, and resulted in effective learning and unlearning abilities simultaneously.’'*

---

> ### Comment · Reviewer_iYfp · 2024-11-21
> **Comments after Author's Rebuttal**
>
> I appreciate the effort from the authors in addressing my concerns. I have some further questions:
>
> - Thanks for providing the time consumption statistics. It seems that PALL indeed suffers from the same efficiency issue as WSN. I wonder if this issue deteriorates for the longer the task length. In other words, it seems that most of the time consumption comes from the binary mask consolidation and gradient masking process. I am not sure if this problem is more severe at the later stages within a long task sequence (e.g., 100 tasks). It would be better if the authors could plot a figure to show the time consumption along the task number increases.
>
> - Thanks for providing the experimental results under the 100-task setting. I have two further questions regarding the experiments: (1) how about the performance compared to other baselines? (2) how about the parameter size increment along the task number increases during this process? It would be better if this could be quantitatively depicted (maybe in the figure), and that is my concrete concern about the scalability compared to simply discussing the performances.
>
> I understand the additional experiments can take some time. The authors can answer my questions according to your time and effort.

---

> > ### Author Response · Authors · 2024-11-24
> > **Response to Further Comments from Reviewer iYfp**
> >
> > We thank the reviewer for their response and appreciation of our efforts during the rebuttal!
> >
> > We provide further clarifications on the scalability experiments considering the raised comments below:
> >
> > >_I am not sure if this problem is more severe at the later stages within a long task sequence (e.g., 100 tasks). It would be better if the authors could plot a figure to show the time consumption along the task number increases._
> >
> > We now **added Figure A1** in Appendix A.3.4 on training time duration comparisons within a longer lifelong scenario. Since we cannot include the figure here, we kindly ask the reviewer to refer to **Figure A1** from page 6 of our revised supplementary materials PDF.
> >
> > We observed that the training time overhead for PALL remains stable as the number of tasks increase. The main overhead in PALL is caused by the parameter gradient masking, which is implemented based on the _cumulative binary mask_, $\mathbf{M}_i$. As the number of tasks grow, we update this cumulative mask in one step via: $\mathbf{M}_i=\bigvee\{\mathbf{m}_j\}$ across all active tasks $j$, then apply it to the gradients. Since this series of logical OR operations do not influence the execution time drastically, we do not see an impactful change for PALL as the number of tasks grow. Thus, this problem does not get more severe for longer task sequences.
> >
> > >(1) how about the performance compared to other baselines?
> >
> > We now performed these experiments on S-TinyImageNet ($T=100$) and **updated Table A6** of Appendix A.3.3, with the competitive baseline methods. Results are also provided below in the form of $\mathcal{A}_l / \mathcal{F}_u$.
> >
> > Our evaluations on how PALL performs in longer scenarios follow the same trend from Table 1 of the main text. We observe that for such longer scenarios, PALL performs favorably in comparison to the baselines with a reasonable trade-off in performance, _when all evaluation criteria are considered together_, i.e., _by ensuring exact unlearning guarantees with memory-efficiency_. Specifically, although PALL yields lower $\mathcal{A}_l$ than WSN as $N_u$ increases, our approach can ensure exact unlearning by design. On the other hand, although training Independent models can also ensure exact unlearning, it becomes a significantly more memory-demanding approach in this longer scenario (further discussed in response to the next question).
> >
> > We now elaborate this better in detail under Appendix A.3.3.
> >
> > |                           | Exact Unl. | $N_u=3$    | $N_u=5$    | $N_u=10$   | $N_u=20$   | $N_u=50$   |
> > |---------------------------|------------|------------|------------|------------|------------|------------|
> > | Sequential                | &#x2716;   | 52.46/0.0  | 51.86/0.0  | 53.01/0.0  | 54.15/0.0  | 51.91/0.0  |
> > | EWC                       | &#x2716;   | 79.53/0.0  | 79.57/0.0  | 79.53/0.0  | 79.59/0.0  | 79.17/0.0  |
> > | DER++                     | &#x2716;   | 76.58/0.45 | 75.04/0.16 | 75.63/0.36 | 74.59/0.45 | 73.22/0.22 |
> > | PackNet                   | &#x2716;   | 81.73/0.0  | 81.76/0.0  | 81.85/0.0  | 81.97/0.0  | 81.75/0.0  |
> > | WSN                       | &#x2716;   | 88.05/0.0  | 87.97/0.0  | 87.62/0.0  | 88.02/0.0  | 87.74/0.0  |
> > | **PALL ($\alpha=0.01$)**  | &#x2714;   | 85.89/0.53 | 85.45/0.41 | 85.18/0.27 | 84.21/0.21 | 81.36/0.17 |
> > | **PALL ($\alpha=0.025$)** | &#x2714;   | 86.11/0.43 | 85.84/0.31 | 85.52/0.22 | 84.29/0.19 | 79.77/0.20 |
> > | **PALL ($\alpha=0.05$)**  | &#x2714;   | 85.80/0.35 | 85.60/0.29 | 85.04/0.23 | 84.0/0.20  | 77.96/0.24 |
> > | Independent               | &#x2714;   | 86.80/0.0  | 86.51/0.0  | 86.53/0.0  | 86.71/0.0  | 86.93/0.0  |
> >
> > >(2) how about the parameter size increment along the task number increases during this process?
> >
> > We have discussed model size requirements of PALL under Section 5.1 of the main text. There, we denote the model size for PackNet, WSN and PALL to be $d+\mathcal{M}_i$, where $d$ indicates the number of parameters (represented in 32-bits) and $\mathcal{M}_i$ consisting of $d$-dimensional boolean (1-bit) masks $\mathbf{m}_j$ for the active tasks.
> >
> > In our S-TinyImageNet ($T=100$) experiments with ViT-T/8 from Table A6, the base model has an inference time size of 20.63 MB for Sequential, EWC, DER++, which remains fixed for any number of tasks. This corresponds to 32-bit representation of approx. 5.4M parameters. As the number of tasks increase to $T=100$, _the model size increment for PALL was observed to be approx. 0.69 MB for each additional task_ (due to the binary mask parameter size increment), which results in a maximum total model size of 89.53 MB at the end of a sequence without unlearning requests. This would be the same for PackNet and WSN. On the other hand, training _Independent_ models would result in a significantly larger model requirement of 2063 MB with $d\times T$ parameters (32-bit), based on _a step-wise increase of 20.63 MB for each new task_.
> >
> > We now added these discussions also under Appendix A.3.3.

---

> ### Comment · Reviewer_iYfp · 2024-11-25
> **Further Comments from Reviewer iYfp**
>
> Thanks for providing additional explanations to my questions. After carefully reading the author's rebuttal, my concerns (i.e., scalability, training efficiency, and model size increment) have been addressed.
>
> Furthermore, I also read the discussion between the authors and other reviewers, and it seems that the questions from my colleagues were also addressed during the peaceful discussions. Thus, I decided to increase my rating to 8 and continue to support the acceptance of this manuscript.

---

> > ### Author Response · Authors · 2024-11-25
> > **Response to Reviewer iYfp**
> >
> > We thank Reviewer iYfp once again for their time and thoughtful suggestions, which have helped improve our manuscript.
> >
> > We are pleased to hear that all concerns are now fully addressed, and also thank the reviewer for re-evaluating their initial score to support the acceptance of our submission - this is greatly appreciated.
> >
> > Best regards,
> >
> >  Authors

---

### Official Review · Reviewer_KbAJ · 2024-10-27

**Soundness:** 3
**Presentation:** 3
**Contribution:** 2
**Rating:** 6
**Confidence:** 4

**Summary:**

This paper proposes privacy-aware lifelong learning (PALL). It is lifelong learning because it learns (explicit) tasks sequentially, and it is privacy-aware because it can unlearn any previously learned task. Tasks are learned on a sparse subset of the parameters only, i.e., the rest are masked out. These sparse active-subset masks are stored for each task, where the sparsity ratio is a hyperparameter. This makes it possible to unlearn a specific task by "removing" the subset determined by its associated mask, where "removing" here means a memory buffer rehearsal on the subsequent tasks of the unlearned task. Specifically speaking, for each subsequent task of the unlearned task (that has not been unlearned so far), we retrain the loss on a memory buffer and additionally regularize the logits to be close to the stored logits in the buffer. This lifelong learning algorith is thus memory-efficient and can preserve privacy by unlearning tasks. Experiments show that this algorithm is sound and performs well in practice.

**Strengths:**

- The paper is well-written
- The proposed method is sound and does well on the experiments.
- Modeling data in terms of tasks is a sound approach.
- Memory-efficiency is a good plus. Learning a subset of parameters per task is practical and performs well on experiments.
- Unlearning is possible and done by memory buffer replay with logits regularization. The performance of unlearning is shown empirically to be exact, which is one of the benefits of PALL.
- A better metric is proposed for measuring the performance of algorithms in sequential task learning-and-unlearning setting. Methods from the literature were adjusted to adapt to this setting, and the authors explained this adaptation well.
- The experiments are extensive and demonstrates the benefits of PALL over other the methods.
- The authors discuss a future direction for PALL to be applicable in class-incremental learning.

**Weaknesses:**

**Main**:
- The algorithm is not necessarily always memory-efficient since the memory complexity still grows linearly in the number of tasks. In other words, the improvement is a constant, albeit a very small one given that it's stored in 1-bit format. However, some models nowadays are trained with less bits (e.g., 8-bits), so storing masks becomes non-trivially expensive when the number of tasks is very large (say, 50).
- The tasks are given explicitly. While this might be the case sometimes in practice (e.g., tasks are user data), it still a disadvantage in terms of privacy (e.g., the data of some task are known to belong to some user).
- Eq. 6. is not novel. It is introduced as a generalized case of DER++ (Buzzega et al., 2020) because the authors say "$\beta=0.5$ yields DER++". Looking at the DER++ paper, it seems to me that their equation is more or less the same with $\beta$ intact ($\beta=0.5$ is the default value). I suggest the author rephrase this section to make this clear since it was written in a way that gives the reader the impression that this is an original contribution that generalizes prior work.
- I would further argue that Eq. 6 is not an elegant remedy for such a privacy-aware algorithm since it requires access to some data from previous tasks.
- The authors mentions "strong privacy considerations, where exact unlearning is possible." The authors should mention that the unlearning is is found to be exact *empirically* because it could be the case that the proposed algorithm would not be able to *unlearn exactly* on real-world data (e.g., foundational models might not be able to unlearn *exactly* even when using PALL).
- There are no investigations done on language data. This is especially important since the right to be forgotten is very applicable in this case.
- Dynamic sparsity does not seem to be very important. Perhaps using static sparsity *and* increasing model size can match the same performance with the same memory cost.
- Some experiments are not directly related to PALL. For example, Table 2. It is quite obvious that having more data in the memory buffer helps (the authors mention a counter case in the text, but I think the difference here is insignificant and the dataset CIFAR-10 is less real-world-like to make conclusions from).

**Minor**:
- Eq. 1., uses inconsistent notation for $\mathcal{L}$ in terms of arguments. Shouldn't contain $\theta_0$ on the left hand side? I also think it would be less convoluted without the $\theta_t \sim$ since it unnecessarily assumes that $\mathcal{L}$ is stochastic.
- Eq. 2., $\mathbf{U}$ and $\mathbf{L}$ not introduced. I believe they are just flag variable?
- Eq. 3., the first constraint is an expression without equality/inequality (min ERM). Perhaps "s.t." should be replaced with an inequality and "min" should be replaced with "argmin".
- I believe experiments and experimental results should traditionally be in the same section.
- $\alpha$ is not clearly introduced, only briefly after Eq. 4. I suggest to reintroduce $\alpha$ somewhere in the experimental results section  for clarity as it is used extensively there.
- The choice $\alpha = 1/T$ provides a fixed memory budget, but this might make comparisons between CIFAR-10 and CIFAR-100 trickier since they have a different number of tasks.
- The experiments are being compared with methods that are not designed for the problem in this work. While this is not necessarily bad, it makes PALL more likely to outperform them since PALL is designed for this problem, so its better peformance, particularly in unlearning, is expected and not surprising.

**Questions:**

- How do you guarantee that a random sparse subset would not have significant overlap with a previous subset?
- Seems like memory buffers are an integral part of this lifelong learning algorithm. Do the authors have an idea how it could be done without them?

---

> ### Author Response · Authors · 2024-11-19
> **Response to Reviewer KbAJ (part 1/2)**
>
> >_**(Q1)** How do you guarantee that a random sparse subset would not have significant overlap with a previous subset?_
>
> We do not constrain the algorithm in terms of the overlap between task-specific subnetwork masks. Our update rule for the importance scores $s_t$ in Eq. (5) is applied to all weights without masking, and any pretrained/frozen weight from previous tasks can be assigned high importance scores if they are found useful to minimize the loss at the current task. If any task $t$ and task $t+1$ to be learned are very similar, the algorithm will ideally optimize a subnetwork mask for task $t+1$ that highly overlaps with the mask for task $t$. If this is not desired, one can disable knowledge transfer and enforce the subnetworks to be independent, which would require masking of the importance score updates, similar to the parameter updates, in Eq. (5).
>
> >_**(Q2)** Seems like memory buffers are an integral part of this lifelong learning algorithm. Do the authors have an idea how it could be done without them?_
>
> The memory buffer is only required to recover performance after unlearning, if knowledge transfer is performed during learning. Currently, no method can jointly satisfy: (1) no catastrophic forgetting, (2) allowing forward knowledge transfer, (3) exact unlearning capabilities, (4) memory-efficiency in terms of the number of model weights. PALL is the first solution at this intersection, and using a memory buffer was a reasonable remedy to balance the desired (2) *knowledge transfer* and (3) *exact unlearning* capabilities, simultaneously. The alternative that excludes memory buffers would be to disable ''(2) *knowledge transfer*'' across tasks, and optimize sparse _independent_ subnetworks (as presented in our Section 5.3), such that any unlearning request could be performed by only resetting the corresponding subnetwork weights.
>
> We now clarified this distinction in our Section 5.3 revisions.
>
> >_**(W1)** The algorithm is not necessarily always memory-efficient since the memory complexity still grows linearly in the number of tasks. In other words, the improvement is a constant, albeit a very small one given that it's stored in 1-bit format. [...]_
>
> Our memory-efficiency claims are only addressed in comparison to training *Independent* models for each task. This was the only existing solution to the problem of *lifelong learning with exact unlearning*. Now, to achieve this, one can use PALL in a more memory efficient way as the number of tasks grow, even in the case of lower weight quantization. This is because of the required total memory usage of training *Independent* models. We re-stated this again in our revisions.
>
> >_**(W2)** The tasks are given explicitly. While this might be the case sometimes in practice (e.g., tasks are user data), it still a disadvantage in terms of privacy (e.g., the data of some task are known to belong to some user)._
>
> This is not a weakness of our method, but instead the nature of the task-incremental lifelong learning problem. We focus on traditional task-incremental lifelong learning scenarios, and in this case the task ID is always provided to the model. Particularly in this context, we are proposing a solution that can ensure exact task unlearning.
>
> >_**(W3)** Eq. 6. is not novel. It is introduced as a generalized case of DER++ (Buzzega et al., 2020) [...] I suggest the author rephrase this section [...]_
>
> We now rephrased our narrative to be more explicit. Indeed, we did not need to re-introduce a novel experience replay objective since existing methods performed well for such a sparse and fast retraining phase. Eq. (6) is generic to apply any experience replay objective by changing the weighting factor. Our formulation of PALL uses the state-of-the-art approach with DER++ ($\beta=0.5$) in the main results, although our further analyses in Table A4 of the Appendix show that DER++ is _not required_ for PALL to perform well.
>
> >_**(W4)** I would further argue that Eq. 6 is not an elegant remedy for such a privacy-aware algorithm since it requires access to some data from previous tasks._
>
> We propose a previously non-existing solution to the problem of ensuring exact unlearning, when lifelong learning *with knowledge transfer* is performed. Therefore, Eq. (6) or a similar experience replay mechanism, is only necessary to recover performance degradations caused by unlearning, *if knowledge transfer is desired*. If forward knowledge transfer is not of concern, a memory replay mechanism or Eq. (6) is also not necessary. Thus, our solution is a generalized formulation that can satisfy all pillars of lifelong learning and exact unlearning.

---

> ### Author Response · Authors · 2024-11-19
> **Response to Reviewer KbAJ (part 2/2)**
>
> >_**(W5)** The authors mentions ''strong privacy considerations, where exact unlearning is possible.'' The authors should mention that the unlearning is found to be exact empirically because it could be the case that the proposed algorithm would not be able to unlearn exactly on real-world data (e.g., foundational models might not be able to unlearn exactly even when using PALL)._
>
> The algorithm does not yield exact unlearning only empirically, but guarantees this by design of the algorithm. Our method resets the subnetwork that observes any task-specific data, and that the new model becomes identical to one that has never observed any data from the task that is unlearned. Considering the reviewer's example, if a foundational model was trained in a task-incremental setting with our proposed subnetwork structures, then PALL could also exactly unlearn a specific task dataset.
>
> >_**(W6)** There are no investigations done on language data. This is especially important since the right to be forgotten is very applicable in this case._
>
> This is not a limitation of our method. Our solution is agnostic to the machine learning task and architecture, and we only focused our experiments to the vision domain due to our interest. We acknowledge the potential in extending our work to language processing tasks, and leave these explorations for future work. We added this in the Discussion section.
>
> >_**(W7)** Dynamic sparsity does not seem to be very important. Perhaps using static sparsity and increasing model size can match the same performance with the same memory cost._
>
> In a true lifelong learning scenario where the number of tasks are unknown and can be very large, dynamically exploring both the sparse connectivity structure and the selective weight sharing can be useful, as opposed to using independent models with static sparsity without knowledge transfer. Exploitation of forward knowledge transfer across related tasks is one of the main pillars in lifelong learning algorithms, and dynamic sparsity helps us to achieve this.
>
> >_**(W8)** Some experiments are not directly related to PALL. For example, Table 2. [...]_
>
> We moved this table to Appendix A.3.1 as Table A1, and edited the text around it by referring to Appendix A.3.1.
>
> >_**(W-Mi1)** Eq. 1, uses inconsistent notation for $\mathcal{L}$ in terms of arguments. [...]_
>
> We corrected the notation in Eq. (1) by fixing the $\theta^0$. The learning algorithm $\mathcal{L}$ is indeed stochastic, and we actually perform lifelong learning sequentially via $\theta^t\sim\mathcal{L}(\theta^{t-1},\mathcal{D}^t)$. Due to this notation we defined early on in Section 3.1, we keep $\theta^t\sim$ in Eq. (1).
>
> >_**(W-Mi2)** Eq. 2., $\mathbf{U}$ and $\mathbf{L}$ not introduced. I believe they are just flag variable?_
>
> Yes, we now defined these terms explicitly.
>
> >_**(W-Mi3)** Eq. 3., the first constraint is an expression without equality/inequality (min ERM). Perhaps "s.t." should be replaced with an inequality and "min" should be replaced with "argmin"._
>
> The current expression is also correct. The first constraint of this expression indicates the overarching lifelong learning goal, that the learning algorithm $\mathcal{L}$ updates the parameters such that the *test set* error is minimized on all previously observed tasks.
>
> >_**(W-Mi4)** I believe experiments and experimental results should traditionally be in the same section._
>
> We renamed Section 3 as ''Experimental Setup'' to avoid the confusion.
>
> >_**(W-Mi5)** $\alpha$ is not clearly introduced, only briefly after Eq. 4. I suggest to reintroduce $\alpha$ somewhere in the experimental results section for clarity as it is used extensively there._
>
> We now reintroduced $\alpha$ again in the caption of Table 1, while discussing the results.
>
> >_**(W-Mi6)** The choice $\alpha=1/T$ provides a fixed memory budget, but this might make comparisons between CIFAR-10 and CIFAR-100 trickier since they have a different number of tasks._
>
> Correct. We tried not to make comparisons of results between these datasets. This is only an intuitive choice to simplify further hyperparameter decisions. We allocate a subnetwork with $1/T$ portion of the complete architecture for each task.
>
> >_**(W-Mi7)** The experiments are being compared with methods that are not designed for the problem in this work. While this is not necessarily bad, it makes PALL more likely to outperform them since PALL is designed for this problem, so its better peformance, particularly in unlearning, is expected and not surprising._
>
> This is not a weakness of PALL, but the nature of the state-of-the-art. We tackle a problem without any existing solution. We needed to create a set of benchmark comparisons from the lifelong learning literature to simply demonstrate why existing lifelong learning methods would not work well in this problem. The only baseline method that we truly compare against is training *Independent* models for each task, which can ensure exact unlearning.

---

> > ### Comment · Reviewer_KbAJ · 2024-11-23
> > **Response to Authors' Rebuttal**
> >
> > I would like to thank the authors for their extensive effort in writing their rebuttal. The authors have clarified some points that I misundertood, and made it clear how their work is positioned in the literature and why their contributions are important. The changes are reflected in the revised version, including extra results that address the other reviewers' concerns. Thus, I now tend to accept this work.

---

> > > ### Author Response · Authors · 2024-11-24
> > > **Response to Reviewer KbAj**
> > >
> > > We thank Reviewer KbAj for their time and thoughtful suggestions, which have helped improve our manuscript.
> > > We are pleased to hear that all concerns are now fully addressed during our rebuttal and revisions.
> > >
> > > We also thank the reviewer for re-evaluating their initial review score - this is greatly appreciated.
> > >
> > > Best regards,
> > >
> > >  Authors

---

### Official Review · Reviewer_ggGQ · 2024-11-01

**Soundness:** 3
**Presentation:** 3
**Contribution:** 3
**Rating:** 6
**Confidence:** 4

**Summary:**

This paper studies the important problem of “privacy-aware life-long learning” where a model must learn from a sequence of tasks, while supporting unlearning of a task previously-encountered in the sequence, when requested. Task unlearning is defined as a modification to the model to make it indistinguishable from one that was trained on the remainder of the sequence, excluding the task to be unlearned. Overall, this problem formulation inherits several desiderata from standard life long learning, e.g. enabling forward knowledge transfer, preventing backwards interference (reduced accuracy on old tasks due to learning new tasks), while being as memory efficient as possible. In addition, the desire to support task unlearning requests brings additional considerations, like being able to do unlearning in an “exact manner” (guaranteeing that influence from the unlearned task is indeed fully removed), with minimal impact to performance of other tasks.

The authors propose a new method to address this problem based on an architecture that can be seen as comprising multiple sub-networks. Each task is assigned a mask (based on learnable parameters determining the importance of each parameter in the model for that particular task), and only modifies the parameters indexed by that mask during learning. Each new learning task in the sequence can only choose to modify previously-unused parameters. To unlearn a task, the parameters that had been affected by that task get reset and a light-weight retraining is performed on only the reset parameters, in order to restore performance for other tasks that might have come after the unlearned task in the sequence. This light-weight retraining is done using a buffer of some randomly-chosen data stored from each task seen so far.

The authors conduct experiments against several baselines from life-long learning and on various datasets. They also conduct thorough ablations to understand the effect of the frequency of unlearning tasks, the effect of the number of finetuning steps, as well as comparisons to an ablation of their system that maintains independent subnetworks within the model but does not facilitate forward transfer.

**Strengths:**

- The paper studies an important problem and presents a novel solution. To the best of my knowledge, this work is the first to provide an exact unlearning solution in a lifelong setting of interleaved learned and unlearned tasks.

- The paper is for the most part well-written and easy to follow.

- The experimental evaluation covers various relevant baselines and ablations.

- The proposed method performs well simultaneously in terms of several metrics and desiderata compared to the baselines considered.

**Weaknesses:**

- It seems that the “independent subnetworks without knowledge transfer” ablation actually performs very similarly to the proposed approach (in Table 3), making it harder to motivate the significantly-more-complex variant for the additional 1% accuracy? Especially given that no confidence intervals are reported there, it’s hard to tell if these differences are significant and whether they justify the additional complexity. Are there perhaps other sequences of tasks / datasets where the “knowledge transfer” would be more “needed”, perhaps making this not the best “benchmark” to showcase the added benefits of knowledge transfer?

- Some clarity issues when describing the method, metrics and setup. Specifically, A) I don’t understand the difference between m_t and \bar m_t. The authors say the latter are the parameters specifically trained using data from D_t, but I don’t understand this statement. What does “specifically trained” mean? B) When describing the metric A_t, it’s unclear whether the final model checkpoint (after addressing the whole sequence) is used to compute accuracy on all tasks, or is a separate checkpoint (right after training on the particular task) used to evaluate the accuracy on each task? The text made me think the former but the notation a_{i,t} and explanation below made me think the latter. C) ““Regularization-based methods [...] were indifferent to task unlearning instructions, since the original methods are not adapted to unlearning” – how were these applied to unlearning tasks, then?

- Insufficient description (in the main paper) for how various other methods were adapted from the standard lifelong setting to the privacy-aware lifelong setting. It is difficult to fully grok how this adaptation was done and why in that particular way (rather than applying other types of approximate unlearning from the literature, for example).

- It is unclear how one should interpret the A_u metric. Having a network perform at chance level in terms of accuracy does not imply an unlearning guarantee, as the authors also acknowledged. It is not obvious that lower A_u is better (an accuracy that is “too low” can compromise privacy and cause vulnerability to attacks). Further, even when taking the right reference point (for how low accuracy should be) into consideration, accuracy-based metrics correlate poorly with more rigorous metrics for unlearning (see e.g. [1]). The authors seem to acknowledge this but still use this metric in the results and in discussions. It would be great to clarify in what way this information should be used and taken into consideration, given this caveat.

- Missing an important baseline: training sequentially with Differential Privacy (DP). This would yield approximate (but theoretically sufficient) unlearning without the need for resetting and retraining any weights. Various DP baselines can be considered: a DP variant of “Sequential”, where all parameters are updated for all tasks, as well as a DP variant of a sub-networks architecture, like in the proposed approach, where each task gets to choose which parameters to update, but in this case addressing unlearning requests can be done trivially (via a “no op”). Depending on the desired level of privacy, I expect such methods to have lower accuracy, but this feels like an important and relevant baseline to include and capture these trade-offs.

- Missing baselines from the unlearning literature. A plethora of approximate unlearning methods have been proposed (see e.g. [2-4]), that try to post-hoc modify a model to remove the influence of a subset of its training data. It feels like these would be readily combinable with the “Sequential” approach, for instance.

- Building further on the above remarks, it would be very useful to build a comparison between the proposed method (and other exact solutions) with inexact unlearning. While the authors have considered some inexact methods, there aren’t proper metrics considered to empirically estimate the privacy of such methods. The only metric considered for this is the accuracy on unlearned tasks, which is a weak notion of privacy, as discussed above. I fully understand that adding empirical privacy auditing is a substantial amount of additional work and potentially difficult to do all of this within a single paper, but I do view this as a clear piece that is missing, or area for future work.

- Another baseline can be training a model sequentially (without subnetworks), and when unlearning is needed, simply retrain the entire model from scratch (using the data in the buffers). I don’t expect this to perform well in terms of accuracy (and would also be less efficient at handling unlearning requests), but would be a useful data point for comparison; this is more akin to the “retrain from scratch” reference point in the unlearning literature. One may additionally consider an “oracle” version of this, that assumes access to all past data as another reference point (I understand that past data is assumed not to be available in life-long learning, hence referring to this as an “oracle”).

- A limitation of this framework is that “user privacy” can only be addressed (via exact unlearning) if a user’s data is cleanly separated in terms of “tasks”. Other scenarios, where all users participated in all “tasks”, would make this framework unable to support users to request their data to be deleted. It would be great to discuss this.

References

[1] Are we making progress in unlearning? Findings from the first NeurIPS unlearning competition. Triantafillou et al.

[2] Eternal sunshine of the spotless net: Selective forgetting in deep networks. Golatkar et al.

[3] Towards unbounded machine unlearning. Kurmanji et al. NeurIPS 2023.

[4] SalUn: Empowering machine unlearning via gradient-based weight saliency in both image classification and generation. Fan et al. ICLR 2024.

**Questions:**

- For the “independent subnetworks without knowledge transfer” ablation, could you explain specifically in what way(s) “Dynamic Sparse Ind.” differs from PALL? It seems both have the learned mask for which parameters to update per task. The authors claim that PALL additionally supports knowledge transfer but it’s not obvious to me what the mechanism is for that, given that each new task only modifies (a chosen subset of) previously-unused weights? I previously assumed that the way that knowledge transfer is realized is that the other weights, even if not directly optimized by the specific task, are still used in the forward pass and therefore still influence the model applied to this task too. Is this correct? And how is this done differently in the “Dynamic Sparse Ind.” variation? Or is there some other difference between the two?

- What is the worst-case sequence of interleaved learning and unlearning requests, w.r.t each metric considered here? Are there cases where the proposed method is no better than naive baselines? It would be great to discuss this in the paper.

---

> ### Author Response · Authors · 2024-11-19
> **Response to Reviewer ggGQ (part 1/3)**
>
> >_**(Q1)** For the ''independent subnetworks without knowledge transfer'' ablation, could you explain specifically in what way(s) ''Dynamic Sparse Ind.'' differs from PALL? [...] The authors claim that PALL additionally supports knowledge transfer but it’s not obvious to me what the mechanism is for that, given that each new task only modifies (a chosen subset of) previously-unused weights?_
>
> The only difference in PALL is the presence of knowledge transfer, by allowing to *selectively* re-use pretrained/frozen weights from previous task-specific subnetworks. This is achieved by optimizing the importance scores $s_t$ over *all* connections via Eq. (5), *without masking*. This means that if any pretrained weight from previous tasks is found useful to solve the current task $t$, its importance score can get higher and this pretrained weight can be selected within $m_t$, resulting in selective re-using of weights. Therefore, in PALL, the dynamically optimized sparse subnetworks can be partially overlapping as a result of the optimization process.
>
> In ''Dynamic Sparse Ind.'', this knowledge transfer via weight-sharing is disabled, thus the dynamically optimized sparse subnetworks are kept *independent*. This is implemented by keeping the importance scores corresponding to any pretrained/frozen weight as 0, such that they are not optimized for selection and cannot be re-used in other tasks.
>
> We restated the main difference in these methods and revised Section 5.3.
>
> >_**(Q2)** What is the worst-case sequence of interleaved learning and unlearning requests, w.r.t each metric considered here? Are there cases where the proposed method is no better than naive baselines? It would be great to discuss this in the paper._
>
> We retrieved these results and added as **Table A7** in the new Appendix A.3.5 subsection.
>
> Results are presented below, in the form of $\mathcal{A}_l^{\text{min}}\uparrow/\mathcal{F}_u^{\text{max}}\downarrow$, where $\mathcal{A}_l^{\text{min}}$ denotes the worst-case task learning performance across randomized repetitions of different user request sequences, and $\mathcal{F}_u^{\text{max}}$ denotes the maximum negative impact of any unlearning request. We do not discuss any other evaluation metrics, since PALL is always superior without catastrophic forgetting ($\mathcal{F}_l=0$) and exact unlearning ($\mathcal{A}_u$ is irrelevant).  We mainly observed that the worst-case evaluations are consistent with the averaged metrics presented in the Table 1 of the main text, and discussed these in Appendix A.3.5.
>
> |                         |  S-CIFAR10 |  S-CIFAR100 | S-TinyImageNet |
> |-------------------------|:----------:|:-----------:|:--------------:|
> | Sequential              |  36.78/0.0 |  21.03/0.0  |    13.93/0.0   |
> | EWC                     |  28.60/0.0 |  51.79/0.0  |    52.18/0.0   |
> | LwF                     |  83.43/0.0 |  48.83/0.0  |    41.04/0.0   |
> | LSF                     | 81.50/3.13 |  51.53/8.87 |   35.42/11.60  |
> | GEM                     | 79.85/5.71 | 52.10/10.28 |   39.52/12.94  |
> | ER                      | 80.63/5.23 | 49.87/15.07 |   39.79/12.12  |
> | DER++                   | 84.45/2.22 |  61.57/4.58 |   44.46/7.11   |
> | PackNet                 |  89.85/0.0 |  72.99/0.0  |    58.35/0.0   |
> | WSN                     |  86.75/0.0 |  69.93/0.0  |    61.34/0.0   |
> | **PALL ($\alpha=1/T$)** | 89.95/1.94 |  69.83/2.17 |   59.91/4.42   |
> | Independent             |  90.85/0.0 |  69.46/0.0  |    60.29/0.0   |
>
> >_**(W1)** It seems that the ''independent subnetworks without knowledge transfer'' ablation actually performs very similarly to the proposed approach [...] Are there perhaps other sequences of tasks / datasets where the ''knowledge transfer'' would be more ''needed'' [...] ?_
>
> We revised these ablation results with additional experiments on **S-TinyImageNet (100 tasks)**. Results are presented below as $\mathcal{A}_l\uparrow/\mathcal{F}_u\downarrow$, which are also presented in our revised **Section 5.3**.
>
> In these new experiments, we test the utility of knowledge transfer across a larger number of tasks, where the model is split into 100 subnetworks. As a result, we were able to observe a larger relative increase in performance as opposed to using independent subnetworks. Here, our main claim is to show that PALL is favorable in lifelong learning and unlearning scenarios where the maximum number of task can be even higher, and unknown a priori.
>
> |                           | $T=40$     | $T=100$    |
> |:-------------------------:|------------|------------|
> |    Static Sparse (Ind.)   | 70.30/0.0  | 80.70/0.0  |
> |   Dynamic Sparse (Ind.)   | 71.19/0.0  | 83.75/0.0  |
> |  **PALL ($\alpha=0.01$)** | 71.00/0.56 | 85.89/0.53 |
> | **PALL ($\alpha=0.025$)** | 72.07/0.36 | 86.11/0.43 |
> |  **PALL ($\alpha=0.05$)** | 72.03/0.32 | 85.80/0.35 |
> |        Independent        | 71.76/0.0  | 86.80/0.0  |

---

> ### Author Response · Authors · 2024-11-19
> **Response to Reviewer ggGQ (part 2/3)**
>
> >_**(W2-A)** I don’t understand the difference between $m_t$ and $\overline{m_t}$._
>
> $\overline{m}_t$ is the binary mask that indicates whether a parameter was updated based on gradients computed with samples from $\mathcal{D}_t$. This $\overline{m}_t$ mask is not necessarily identical to the task-specific subnetwork mask $m_t$, because $m_t$ can additionally indicate weights trained on previous tasks due to the use of knowledge transfer via weight-sharing across tasks.
>
> >_**(W2-B)** When describing the metric $A_t$, it’s unclear whether the final model checkpoint (after addressing the whole sequence) is used to compute accuracy on all tasks, or is a separate checkpoint? [...]_
>
> We compute $\mathcal{A}_l$ and $\mathcal{A}_u$ by using the final checkpoint at the end of the lifelong learning and unlearning sequence.
>
> These metrics are defined by using $a_{r,t}$ in Eq. (7), where $r$ is the length of the whole request sequence, and in the text we define: ''$a_{i,t}$ denotes the test set accuracy after request $i$ was completed''.
>
> >_**(W2-C)** ''Regularization-based methods [...] were indifferent to task unlearning instructions, since the original methods are not adapted to unlearning'' – how were these applied to unlearning tasks, then?_
>
> >_**(W3)** Insufficient description (in the main paper) for how various other methods were adapted from the standard lifelong setting to the privacy-aware lifelong setting. [...]_
>
> We clarified these in our revisions. Due to space limitations, we could only include further details of how each method was adapted under ''Unlearning Implementations'' subsection of Appendix A.2. We revised Section 4.2 of the main text to make our pointers to the Appendix more explicit.
>
> For EWC, LwF and Sequential training, upon receiving a task unlearning request, we did not perform any specific parameter update step (i.e., they were indifferent to the task unlearning instruction). In Sequential, we simply ignore the unlearning request. For EWC and LwF, we delete the previously stored model weights/checkpoint from the task to be unlearned. For EWC, we also discard the previously computed and stored Fisher information matrix from the task to be unlearned.
>
> We performed *inexact unlearning* adaptations on lifelong learning algorithms, since it was a seamless integration. Existing approximate unlearning algorithms mostly require access to previously observed data, thus it is harder to adapt these inexact methods to lifelong learning. Nevertheless, these baselines were not critical for the main outcome of our work since our focus was to achieve exact unlearning, which is a stricter notion of privacy.
>
> >_**(W4)** It is unclear how one should interpret the $A_u$ metric. [...]_
>
> We included this metric only to compare the approximate ''unlearning'' of the baseline methods, towards achieving poor classification on unlearned tasks. Indeed, this does not guarantee privacy. We do not make any strong claims based on $\mathcal{A}_u$, since PALL  can achieve exact unlearning regardless of $\mathcal{A}_u$. We now explicitly state this in our revisions under Section 4.2:
>
> *''[...] it is important to note that a lower $\mathcal{A}_u$ does not necessarily correspond to an exact unlearning guarantee. We include $\mathcal{A}_u$ only to evaluate inexact unlearning baselines through a weak measure. We leave detailed investigation of inexact unlearning methods with better metrics, e.g., via empirical privacy auditing (Steinke et al., 2024), for future work.''*
>
> >_**(W5)** Missing an important baseline: training sequentially with Differential Privacy (DP). [...]_
>
> >_**(W9)** A limitation of this framework is that “user privacy” can only be addressed (via exact unlearning) if a user’s data is cleanly separated in terms of “tasks”. Other scenarios, where all users participated in all “tasks”, would make this framework unable to support users to request their data to be deleted. It would be great to discuss this._
>
> We considered differential privacy (DP), but excluded it from the current submission to be concise. DP baselines would help maintaining deletion guarantees *for each data sample of each task* as an extreme case with stricter privacy. We perform exact task unlearning altogether. However, related to the other comment **(W9)**, we could potentially utilize differentially private optimization while learning each task, to preserve privacy of individual data points within tasks.
>
> We now included a brief limitation discussion on DP and selective data unlearning, at the end of our revised paper, in the context of potential future work:
>
> *''Our work also does not yet consider selective data unlearning, but instead performs complete task unlearning.
> To achieve stricter privacy with deletion guarantees for each data sample, our approach can be combined with differentially private optimization methods (Lai et al., 2022), in future work.''*

---

> ### Author Response · Authors · 2024-11-19
> **Response to Reviewer ggGQ (part 3/3)**
>
> >_**(W6)** Missing baselines from the unlearning literature. A plethora of approximate unlearning methods have been proposed (see e.g. [Golatkar et at.; Kurmanji et al.; Fan et al.]) [...]. It feels like these would be readily combinable with the “Sequential” approach, for instance._
>
> These baselines are not applicable to the *Sequential* approach, since they require access to previously observed ''retain'' or ''forget'' set samples in their objectives. An important factor to consider here is the lack of access to previous data in lifelong learning. One *can* combine Sequential learning with another inexact unlearning algorithm that does not require such data. However, this would only be another *inexact* baseline. We prefer to keep our main focus on *exact* unlearning (i.e., adaptation of SISA as Independent model training for each task).
>
> >_**(W7)** [...] it would be very useful to build a comparison between the proposed method (and other exact solutions) with inexact unlearning. While the authors have considered some inexact methods, there aren’t proper metrics considered to empirically estimate the privacy of such methods. [...] I fully understand that adding empirical privacy auditing is a substantial amount of additional work and potentially difficult to do all of this within a single paper [...]._
>
> We thank the reviewer for this comment and suggestion for future work. We now mention this missing piece under Section 4.2, where we define our evaluation metrics. We include $\mathcal{A}_u$ only to evaluate inexact unlearning baselines through a weak measure. We leave the detailed investigation of inexact methods with better metrics, e.g., via empirical privacy auditing, for future work.
>
> >_**(W8)** Another baseline can be training a model sequentially (without subnetworks), and when unlearning is needed, simply retrain the entire model from scratch (using the data in the buffers). [...] One may additionally consider an ``oracle'' version of this, that assumes access to all past data as another reference point [...]_
>
> We implemented this on S-CIFAR, annotated as ''ER (Retraining)'', in the new **Table A5** provided below.
>
> Since the suggested setting requires a memory buffer, we perform experience replay (ER) based task-incremental learning as well. For unlearning, we retrain the entire model from scratch using the memory buffer. We exclude the ''oracle'' setting with access to all past data, since it would conflict with our storyline from a traditional continual learning perspective. ''ER (Retraining)'' results showed that the performance was highly restricted by the memory buffer following the first unlearning request. Although retraining the model yields exact unlearning, re-learning previous tasks from scratch using ER did not scale well even to the slightly more complex task of S-CIFAR100.
>
> |                         |                         |                           |                           |                           |   |   |                         |                         |                           |                           |                           |
> |-------------------------|-------------------------|---------------------------|---------------------------|---------------------------|---|---|-------------------------|-------------------------|---------------------------|---------------------------|---------------------------|
> | **S-CIFAR10 ($T=5$)**   | $\mathcal{A}_l\uparrow$ | $\mathcal{A}_u\downarrow$ | $\mathcal{F}_l\downarrow$ | $\mathcal{F}_u\downarrow$ |   |   | **S-CIFAR100 ($T=10$)** | $\mathcal{A}_l\uparrow$ | $\mathcal{A}_u\downarrow$ | $\mathcal{F}_l\downarrow$ | $\mathcal{F}_u\downarrow$ |
> | Independent             | 95.19                   | **Exact**                 | **0.0**                   | 0.0                       |   |   | Independent             | 73.22                   | **Exact**                 | **0.0**                   | 0.0                       |
> | **PALL ($\alpha=1/T$)** | 94.34                   | **Exact**                 | **0.0**                   | 0.60                      |   |   | **PALL ($\alpha=1/T$)** | 72.35                   | **Exact**                 | **0.0**                   | 0.40                      |
> | ER (Retraining)         | 78.80                   | **Exact**                 | 2.05                      | 7.53                      |   |   | ER (Retraining)         | 34.50                   | **Exact**                 | 4.07                      | 20.40                     |
> | ER                      | 87.88                   | 58.48                     | 3.45                      | 1.61                      |   |   | ER                      | 57.63                   | 42.69                     | 4.67                      | 7.91                      |

---

> > ### Comment · Reviewer_ggGQ · 2024-11-22
> > **response to authors**
> >
> > Dear authors,
> >
> > Thank you for your thorough responses, additional experiments and related modifications to the paper.
> >
> > Your clarifications (re: knowledge transfer, and relatedly, \bar m_t, adaptations of classic methods to this setting, as well as other more minor points), discussion on connections with inexact unlearning, the accuracy metric, connections with DP and when one might use that approach, and directions for future work address those concerns fully.
> >
> > Thank you for the additional results on S-TinyImageNet, it's nice to see that indeed on this more challenging problem PALL outperforms the ablations by a larger margin. Similarly, thanks for adding the ER baseline, it's great to see that those results confirm our intuition that that method does not scale well and is outperformed by PALL.

---

> > > ### Author Response · Authors · 2024-11-24
> > > **Response to Reviewer ggGQ**
> > >
> > > We thank Reviewer ggGQ for their time and thoughtful suggestions, which have helped improve our manuscript. We are pleased to hear that all concerns are now fully addressed during our rebuttal and revisions.
> > >
> > > We would also greatly appreciate if the reviewer considers re-evaluating their initial borderline rating for our submission, as our work has received good scores across all criteria: soundness, contribution, and presentation. We would be also happy to address any remaining concerns regarding our submission.
> > >
> > > Best regards,
> > >
> > >  Authors

---

### Official Review · Reviewer_QDPb · 2024-11-10

**Soundness:** 2
**Presentation:** 2
**Contribution:** 2
**Rating:** 5
**Confidence:** 4

**Summary:**

This paper formulates a task-incremental learning and unlearning problem with strong privacy considerations, by extending the traditional task-incremental learning setup to allow exact task unlearning instructions. The authors present privacy-aware lifelong learning (PALL) as a memory-efficient algorithmic solution to this new problem.

**Strengths:**

1. A novel experimental setup is proposed to combine lifelong learning and machine unlearning, addressing key challenges in both domains, which have not been addressed to date.
2. This paper presents privacy-aware lifelong learning (PALL) as a memory-efficient algorithmic solution to this setup, which enables learning without catastrophic forgetting, allows learnable forward knowledge transfer, and ensures exact unlearning guarantees by design.
3. The algorithmic empirically demonstrates the scalability of PALL on both convolutional benchmark architectures and attention-based vision transformers, yielding a stable performance in highly dynamic lifelong learning scenarios with randomly arriving unlearning requests.

**Weaknesses:**

1. This paper presents a new problem setting, but this problem lack significant innovation compared with existing problems. What is the most different points that distinguish this problem from others? It seems the objective of this problem in section 3.2 also holds for existing problem such as domain incremental learning and unlearning. Is there any challenges specific to this problem so that we must formulate it as a new problem?
2. The methods presented for the problem setting proposed in this paper appear to be well-established and lack significant innovation. The authors should either highlight any novel contributions or improvements to existing methods or explore more advanced techniques that could offer better solutions to the unique aspects of the problem.
3. This paper introduces an interesting approach, but it is unclear how the proposed method has been specifically tailored to address the new problem setting. The authors should provide a more detailed explanation of how their approach was designed to handle the new settings.
4. The paper should clarify why the task-incremental setting was chosen instead of class- or domain-incremental learning. Providing a rationale for this choice would help to better understand the appropriateness of the selected setting for the problem at hand.
5. The cited references of Regularization-based methods are mostly foundational works, but some are relatively dated, potentially overlooking recent improvements or newer approaches in the field.
6. The details of the Baseline you compared are not clearly described, e.g., how you tailor these baseline methods to fit the proposed problem setting?
7. In the experimental section, the results should be compared with the latest state-of-the-art (SOTA) models in Continual Learning. A more detailed comparison with existing SOTA methods would provide a clearer context for evaluating the proposed approach and strengthen the overall analysis.
8. In the section on Comparisons to Independent Subnetworks without Knowledge Transfer, the descriptions of the two configurations—static sparsity and dynamic sparsity—are not sufficiently clear. Although static sparsity is introduced as a random initialization of T independent sparse subnetworks, and dynamic sparsity as an optimization of sparse connectivity via score optimization, the explanation lacks details on how each configuration functions within the broader experimental setup.

**Questions:**

1. The cited references for Regularization-based methods mainly cover foundational works, but the discussion could benefit from including more recent studies that reflect the latest advancements in this area.
2. The experience settings of other baselines are different from those in this paper. The Baseline Methods section should describe more detailed setups.
3. Including SOTA models would strengthen the study’s impact by showcasing its performance relative to the best-performing approaches in the field.
4. It would be helpful to clarify the initialization and optimization processes more explicitly and outline each configuration's practical implications. Adding these clarifications would aid in understanding the differences and rationale behind these two configurations, improving the overall readability and transparency of this section."

---

> ### Author Response · Authors · 2024-11-19
> **Response to Reviewer QDPb (part 1/2)**
>
> >_**(Q1)** The cited references for Regularization-based methods mainly cover foundational works, but the discussion could benefit from including more recent studies that reflect the latest advancements in this area._
>
> >_**(W5)** The cited references of Regularization-based methods are mostly foundational works, but some are relatively dated, potentially overlooking recent improvements or newer approaches in the field._
>
> Our work focuses on a completely novel problem setting, which was not explored via regularization-based continual learning methods. We are not aware of any particular regularization-based lifelong learning work that is specifically related to our scope in the context of exact machine unlearning. Perhaps the reviewer could elaborate more on this comment on which particular works could be of interest?
>
> >_**(Q2)** The experience settings of other baselines are different from those in this paper. The Baseline Methods section should describe more detailed setups._
>
> >_**(W6)** The details of the Baseline you compared are not clearly described, e.g., how you tailor these baseline methods to fit the proposed problem setting?_
>
> All implementation details of the baseline methods are discussed in Appendix A.2 under subsection ''Unlearning Implementations''. We now revised Section 4.2 of the main text to make our pointers to the Appendix more explicit.
>
> _**Unlearning Implementations:** For Sequential, EWC and LwF, we do not perform any changes to the model parameters for task unlearning. For EWC and LwF, we only discard the previously stored model weights (which are used for continual learning in these methods), corresponding to the unlearned task. For EWC, we also discard the previously stored Fisher information matrix corresponding to the unlearned task. We perform algorithm-specific unlearning updates for LSF, gradient episodic memory (GEM), experience replay (ER), dark experience replay (DER++), when task unlearning is instructed. For task unlearning with GEM, ER and DER++, we implement an experience replay finetuning objective on the remaining tasks' episodic memories, and also predict uniform distributions using the unlearned task's episodic memories to accelerate forgetting. Task unlearning with LSF exploits its unique notion of manipulating the input space by superposing distinct auxiliary mnemonic codes on task-specific data. For unlearning, the mnemonic code of the unlearned task is directly provided to the network to be predicted incorrectly via a uniform label distribution, and the remaining task codes are directly provided to the network to be predicted accurately._
>
> >_**(Q3)** Including SOTA models would strengthen the study’s impact by showcasing its performance relative to the best-performing approaches in the field._
>
> >_**(W7)** In the experimental section, the results should be compared with the latest state-of-the-art (SOTA) models in Continual Learning. A more detailed comparison with existing SOTA methods would provide a clearer context for evaluating the proposed approach and strengthen the overall analysis._
>
> Since our work focuses on a novel lifelong learning problem that involves exact unlearning guarantees, the state-of-the-art (SOTA) in this setting corresponds to training *Independent* models for each task. This was already included in all our analyses as an upper performance baseline as the best-performing approach in the proposed problem of task-incremental lifelong learning with exact unlearning. Additionally, we implemented several continual learning methods with inexact unlearning approximations, as weaker baselines as well.
>
> >_**(Q4)** It would be helpful to clarify the initialization and optimization processes more explicitly and outline each configuration's practical implications. Adding these clarifications would aid in understanding the differences and rationale behind these two configurations, improving the overall readability and transparency of this section._
>
> We have included all initialization and optimization hyperparameter specifications for each method explicitly in Appendix A.2 - ''Training Configurations and Implementations'', under subsections ''Model Training'' and ''Baseline Methods''.

---

> ### Author Response · Authors · 2024-11-19
> **Response to Reviewer QDPb (part 2/2)**
>
> >_**(W1a)** This paper presents a new problem setting, but this problem lack significant innovation compared with existing problems. What is the most different points that distinguish this problem from others? [...]_
>
> >_**(W2)** The methods presented for the problem setting proposed in this paper appear to be well-established and lack significant innovation. The authors should either highlight any novel contributions or improvements to existing methods or explore more advanced techniques that could offer better solutions to the unique aspects of the problem._
>
> In this paper, both the introduced problem formulation and the proposed solution are novel. There are no well-established methods that can tackle the problem of efficient lifelong learning with exact unlearning guarantees. We adapted existing lifelong learning methods in the context of the new problem, and demonstrated how these methods do not generalize to a setting with exact unlearning guarantees.
>
> >_**(W1b)** [...] It seems the objective of this problem in section 3.2 also holds for existing problem such as domain incremental learning and unlearning. Is there any challenges specific to this problem so that we must formulate it as a new problem?_
>
> >_**(W4)** The paper should clarify why the task-incremental setting was chosen instead of class- or domain-incremental learning. Providing a rationale for this choice would help to better understand the appropriateness of the selected setting for the problem at hand._
>
> We are not aware of existing works that studies ''domain incremental learning with exact unlearning''. Our work studies the problem of task-incremental learning with *exact* unlearning, which was not previously explored. Our formulation can, however, naturally extend to domain-incremental learning with *exact* unlearning as well. We have discussed a future direction for PALL to be applicable in class- or domain-incremental learning settings in our manuscript (see last paragraph of the Discussion section). Indeed, PALL is not readily applicable to a class- or domain-incremental scenario, since model disentanglement with task-specific subnetworks based on task IDs are necessary to ensure exact unlearning. To be applicable in such settings, one can use an auxiliary algorithm to first identify the context/task, and subsequently utilize task-specific subnetwork components via gating as previously suggested by various lifelong learning methods [1-2].
>
> Aljundi et al., ''Expert Gate: Lifelong Learning with a Network of Experts'', CVPR 2017.
>
> Oswald et al., ''Continual learning with hypernetworks'' ICLR 2020.
>
> >_**(W3)** This paper introduces an interesting approach, but it is unclear how the proposed method has been specifically tailored to address the new problem setting. The authors should provide a more detailed explanation of how their approach was designed to handle the new settings._
>
> This comment is not very specific, and we hope that the reviewer can elaborate which parts are not clear. Currently, the complete Section 3 describes how PALL is tailored to the problem of achieving exact unlearning in the context of lifelong learning without catastrophic forgetting and allowing forward knowledge transfer, with memory-efficiency.
>
> >_**(W8)** In the section on Comparisons to Independent Subnetworks without Knowledge Transfer, the descriptions of the two configurations—static sparsity and dynamic sparsity—are not sufficiently clear. Although static sparsity is introduced as a random initialization of T independent sparse subnetworks, and dynamic sparsity as an optimization of sparse connectivity via score optimization, the explanation lacks details on how each configuration functions within the broader experimental setup._
>
> We define ''independent static sparsity'' as random initialization of $T$ independent sparse subnetworks with $1/T$ connectivity within a model, where the connectivity structure is kept frozen during training. We define ''independent dynamic sparsity'' to be the same as PALL, without the capability to perform knowledge transfer across tasks such that subnetworks are not overlapping. These models then process the user request sequence similar to the previous *Independent* baseline, following the detailed optimization procedures outlined under Appendix A.2 ''Training Configurations and Implementations''. We revised Section 5.3 to clarify our descriptions better.

---

> > ### Comment · Reviewer_QDPb · 2024-12-03
> >
> > Thanks for the responses. However, the authors did not provide any extension to more recent works for the proposed method, so I keep my rating.

---

> > > ### Author Response · Authors · 2024-12-03
> > > **Response to Reviewer QDPb**
> > >
> > > Thanks to the reviewer for the response.
> > >
> > > >_[...] the authors did not provide any extension to more recent works for the proposed method, so I keep my rating._
> > >
> > > Unfortunately, it remains unclear to us what is specifically implied by _"providing an extension to more recent works for the proposed method"_. We would like to kindly point out that, in our initial rebuttal, we have already sought clarification from the reviewer regarding this point. Below, we quote from our rebuttal response to **(Q1)**:
> > >
> > > >_Our work focuses on a completely novel problem setting, which was not explored via regularization-based continual learning methods. We are not aware of any particular regularization-based lifelong learning work that is specifically related to our scope in the context of exact machine unlearning. **Perhaps the reviewer could elaborate more on this comment on which particular works could be of interest?**_
> > >
> > > In this work, both the introduced problem formulation and the proposed solution are novel. Currently, there are no well-established methods capable of addressing the challenge of efficient lifelong learning while also providing exact unlearning guarantees. We already adapted several state-of-the-art continual learning methods to our problem in the experimental comparisons, and demonstrated how these methods do not generalize to a setting with exact unlearning guarantees. It would have been very helpful if the reviewer had provided more specific details, allowing us to address any questions more effectively.

---

### Author Response · Authors · 2024-11-19
**Rebuttal by Authors**

We thank all reviewers for their time to evaluate our work.

We are pleased that **all reviewers** found our contribution to be *novel* and our manuscript to be *well-written*.
We thank the reviewers for the positive comments on finding our work *to study an important problem for the first time with exact unlearning guarantees* (**Reviewers QDPb, ggGQ & iYfp**), with *significant contributions in terms of memory-efficiency and practicality* of the algorithm (**Reviewers KbAJ & iYfp**), and that the proposed method is *sound* with *thorough experiments* and *strong results* (**Reviewers KbAJ, QDPb, ggGQ**).

During our rebuttal, we performed the following new analyses:
- experiments on S-TinyImageNet (100 tasks) with up to $N_u=50$ unlearning requests,
- benchmarking training time comparisons,
- retrieved the worst-case evaluation metrics corresponding to the averaged results in Table 1 of main text,
- implemented another exact unlearning baseline with model retraining using experience replay.

Accordingly, we performed the following content changes in our submission:
- added Appendix A.3.3 with Table A6 to discuss the scalability of PALL to longer lifelong scenarios, as suggested by **Reviewer iYfp**,
- added Appendix A.3.4 with Table A8 to discuss comparisons of training times, as suggested by **Reviewer iYfp**,
- added Appendix A.3.5 with Table A7 to present the worst-case evaluation metrics, as suggested by **Reviewer ggGQ**,
- added Table A5 under Appendix A.3.2 with another exact unlearning baseline using experience replay, as suggested by **Reviewer ggGQ**,
- revised Section 5.3 with additional results for comparisons to independent subnetworks without knowledge transfer and memory buffers, as suggested by **Reviewers ggGQ and KbAJ**,
- added clarifications on the $\mathcal{A}_u$ metric and how to consider inexact unlearning baselines, as suggested by **Reviewer ggGQ**,
- added pointers to our clarifications on how each baseline method was adapted to unlearning instructions, as suggested by **Reviewers QDPb & ggGQ**,
- revised our discussions on applicability to class-/domain-incremental settings, as suggested by **Reviewers QDPb & iYfp**,
- added discussions on selective unlearning capabilities & differentially private optimization, as suggested by **Reviewer ggGQ**,
- added future work discussions on applicability of PALL to language processing tasks, as suggested by **Reviewer KbAJ**.
- rephrased the narrative on Eq. (6) in relation to existing methods, and added clarifications on the memory-efficiency claims, as suggested by **Reviewer KbAJ**,
- moved the old Table 2 from the main text to the Appendix A.3.1, as mentioned by **Reviewer KbAJ**.

We uploaded a revised version of the main paper and the appendix, with all changes marked in color.

We further address point-by-point reviewer comments below.

---

### Meta-Review · Area_Chair_vbyx · 2024-12-24

**Metareview:**

Summary

The paper explores the challenge of achieving task-incremental lifelong learning while enabling exact task unlearning, both with minimal memory overhead. The proposed method leverages a single neural network where parameters are strategically tracked and adjusted to learning or unlearning objectives, with some parameters shared across tasks. This innovative approach demonstrates strong empirical performance across diverse network architectures.

Strengths

The paper's strengths lie in its innovative approach to lifelong unlearning, tackling the challenge of achieving exact unlearning in a sequential task setting. It introduces a method that facilitates selective knowledge transfer through shared parameters within a single model, effectively balancing retention and unlearning. The novel consideration of the inherently contradictory goals of lifelong learning and unlearning shows the paper's originality. Additionally, the proposed method demonstrates strong empirical performance across various neural network architectures.

Weaknesses

While the proposed approach is effective for task-incremental or domain-incremental learning and removal, it does not generalize to instance-level unlearning, limiting its broader applicability. Related empirical studies are also lacking. The lack of robust baselines, such as sequential training with Differential Privacy, unlearning baselines, or inexact unlearning approaches, limits the comprehensiveness of the evaluation. Finally, the term "user privacy" is overused, as the method addresses only a narrow aspect of privacy through exact task unlearning.

Recommendation

Despite its limitations, the paper offers significant contributions that justify its acceptance. It addresses an important and underexplored challenge by proposing a novel method that integrates the contradictory goals of lifelong learning and (task) unlearning into a single framework. The approach demonstrates strong empirical performance across diverse network architectures. Moreover, exploring exact task unlearning with minimal memory requirements is timely and relevant, contributing to advancements in efficient and privacy-aware machine learning. In contrast, certain aspects, such as tasks, baseline and experimental details, could be improved.

**Additional Comments On Reviewer Discussion:**

During the rebuttal phase, the authors made notable efforts to address reviewer feedback by adding empirical studies, improving discussions, and enhancing the clarity of the paper and supplementary materials.

Reviewer QDPb maintained a score of 5, citing unresolved concerns about the novelty of the method, the rationale for choosing task-incremental settings, and the lack of ablation studies and new baseline comparisons. While the authors explained their choices and deferred extensive new comparisons to future work, the reviewer did not engage further after the initial reply.

Reviewer ggGQ gave a 6, acknowledging improved clarity but emphasizing the need for more extensive experiments and comparisons across diverse unlearning tasks, which the authors deferred to future studies.

Reviewer KbAJ also gave a 6, highlighting the paper's merits in novelty and empirical contributions but raising concerns about memory efficiency and the limited dataset scope. Following clarification, this reviewer leaned toward acceptance.

Reviewer iYfp gave a high score of 8, praising the novel perspective, effective experiments, and improvements in response to concerns. While acknowledging some scalability issues and the focus on task-incremental settings, the additional experiments led the reviewer to fully support acceptance.

Overall, the authors successfully addressed most key concerns or explicitly deferred them to future work, demonstrating sufficient contribution to warrant the paper's acceptance.

---

### Decision · Program_Chairs · 2025-01-22

Accept (Poster)